# Constrained Diffusion for Protein Design with Hard Structural Constraints

**Jacob K. Christopher**
University of Virginia
csk4sr@virginia.edu

**Austin Seamann**
Rutgers University
als515@rutgers.edu

**Jingyi Cui**
University of Virginia
cau8rc@virginia.edu

**Sagar Khare**
Rutgers University
sagar.khare@rutgers.edu

**Ferdinando Fioretto**[*]
University of Virginia
fioretto@virginia.edu

## Abstract

Diffusion models offer a powerful means of capturing the manifold of realistic protein structures, enabling rapid design for protein engineering tasks. However, existing approaches observe critical failure modes when precise constraints are necessary for functional design. To this end, we present a constrained diffusion framework for structure-guided protein design, ensuring strict adherence to functional requirements while maintaining precise stereochemical and geometric feasibility. The approach integrates proximal feasibility updates with ADMM decomposition into the generative process, scaling effectively to the complex constraint sets of this domain. We evaluate on challenging protein design tasks, including motif scaffolding and vacancy-constrained pocket design, while introducing a novel curated benchmark dataset for motif scaffolding in the PDZ domain. Our approach achieves state-of-the-art, providing perfect satisfaction of bonding and geometric constraints with no degradation in structural diversity.

## 1 Introduction

Diffusion models have revolutionized protein engineering with notable successes demonstrated in the design of protein monomers, assemblies, and protein binders against biomolecular targets (Watson et al., 2023). In many cases, predefined binding or catalytic motifs are introduced into designed proteins via motif scaffolding but there are no guarantees that the generated backbones will accurately include the motif (Trippe et al., 2022; Didi et al., 2023). Furthermore, the motifs are typically pre-defined as structural fragments, rather than more physically-based (e.g. hydrogen bonds to chosen target residues), which narrows the accessible design space (Song et al., 2024). Negative space constraints (e.g. tunnels for substrate access and product egress), while a ubiquitous feature of naturally evolved proteins such as enzymes, are not readily incorporated in current generative protein models. These obstacles restrict the scope of design goals accessible to current methods.

These limitations highlight the broader challenge of designing structured objects under strict feasibility. Many existing approaches augment diffusion models through constraint guidance (Ho et al., 2020; Ho & Salimans, 2022). For example, Gruver et al. (2023) inject soft constraints into discrete sequence prediction for protein design through gradient-based guidance; however, while guidance approaches result in increased feasibility rates, they fail to consistently provide constraint adherent outputs. Others adopt post-processing optimizations which more rigorously target the constraint set. However, these methods rely on either simplifications of the highly nonconvex constraint set (e.g., matching an existing ligand template) or result in samples falling outside the data manifold (Bergues et al., 2025; Christopher et al., 2024). A more effective approach is to inject constraint into the generative process, e.g., by projecting intermediate states back to the feasible set (Christopher et al., 2024). Yet projecting noisy states early in the sampling process has been show to disrupt the diffusion trajectory, potentially biasing samples (Blanke et al., 2025).

---

[*]Contact author.

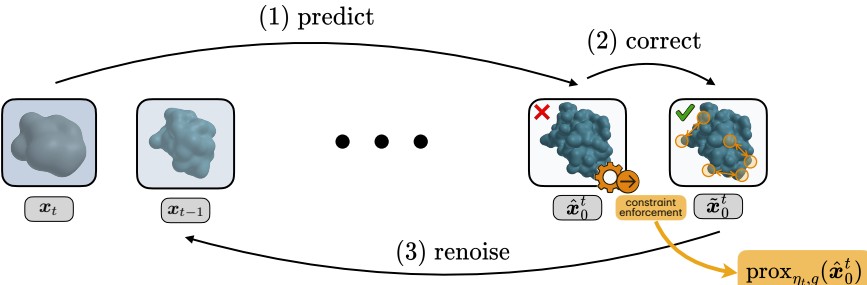

Figure 1: Illustration of our stochastic proximal sampling for structured-constrained protein design.

To resolve this tension, this paper proposes to view constrained diffusion through the lens of stochastic proximal methods. To enable strict constraint enforcement throughout the diffusion process, while removing the need to project at earlier noisy states, we introduce a framework which applies *final-state corrections*. Proximal steps are applied to a predicted clean posterior, rather than on a noisy intermediate state, and the feasible clean state is then renoised to steer the sampling trajectory along the data manifold, while converging to exact feasibility at the terminal state. A schematic illustration of the proposed scheme is provided in Figure 1.

**Contributions.** This paper makes five key contributions: **(1)** It introduces a stochastic proximal method for constrained diffusion feasibility; **(2)** based on this view, it proposes a consensus-based ADMM scheme that separates local stereochemistry from sparse global couplings; **(3)** it provides a theoretical analysis characterizing the convergence to the constraint set and provides arguments for why final-state projection is preferred over per-step projections; **(4)** it demonstrates the efficacy of the approach on challenging protein design tasks with nonconvex constraints, including global topology (e.g., chain closure, ligand binding feasibility) and local stereochemistry (bond lengths, angles, and chirality), achieving perfect constraint satisfaction and providing state-of-the-art performance; **(5)** it introduces a novel curated benchmark for protein motif scaffolding in PDZ domains, providing the first systematic standard for constrained diffusion methods in modular domain engineering.

## 2 SETTINGS AND BACKGROUND

The goal of *de novo* protein design is to generate three-dimensional representations which satisfy physical plausibility and functional requirements, such as protein motif scaffolding, binder design, and monomer generation. This task is fundamentally constrained by the physics and chemistry of proteins, where bond geometries must be preserved, chains must remain connected, and higher-level properties such as specific inter-chain interactions or interface complementarity must be realized.

**Diffusion-based protein backbone generation.** Let $p_{\text{data}}$ denote the unknown distribution of clean molecular structure representations $\boldsymbol{x}_0 \in \mathbb{R}^d$ (e.g., atomic coordinates). Consider a constrained feasibility set $\mathcal{C} \subset \mathbb{R}^d$, encoding physical and geometric constraints, seeking to sample from the target distribution

$$p_{\mathcal{C}}(\boldsymbol{x}_0) \propto p_{\text{data}}(\boldsymbol{x}_0) \, \mathbb{1}\{\boldsymbol{x}_0 \in \mathcal{C}\}, \tag{$\star$}$$

where $\mathbb{1}\{\boldsymbol{x}_0 \in \mathcal{C}\}$ is an indicator function on $\mathcal{C}$. To achieve this, a diffusion model learns to reconstruct samples from the data distribution $p_{\text{data}}$ by coupling a *forward* noising Markov chain with a learned *reverse* denoising chain (Song et al., 2020; Ho et al., 2020).

*Forward process.* Diffusion models construct a Markov chain $\{\boldsymbol{x}_t\}_{t=0}^T$ starting from $\boldsymbol{x}_0 \sim p_{\text{data}}$. At each step, Gaussian noise is added according to a fixed variance schedule $\{\alpha_t\}_{t=0}^T$. This process admits a closed-form marginal distribution, $q(\boldsymbol{x}_t \mid \boldsymbol{x}_0) = \mathcal{N}(\sqrt{\bar{\alpha}_t}\,\boldsymbol{x}_0, \, (1 - \bar{\alpha}_t)I)$, where $\bar{\alpha}_t = \prod_{s=1}^t (1 - \alpha_s)$. As $t \to T$, the distribution converges to an isotropic Gaussian $q(\boldsymbol{x}_T) \approx \mathcal{N}(\mathbf{0}, I)$.

*Reverse process.* A generative model learns to approximate the reverse dynamics, sampling $\boldsymbol{x}_{t-1}$ given $\boldsymbol{x}_t$. Since the true reverse kernel $q(\boldsymbol{x}_{t-1} \mid \boldsymbol{x}_t)$ is intractable, a neural network $\mathbf{x}_\theta(\boldsymbol{x}_t, t)$ is introduced to parameterize this transition. At inference, the learned reverse transitions are applied iteratively, gradually denoising a random Gaussian vector $\boldsymbol{x}_T \sim \mathcal{N}(\mathbf{0}, I)$ into a clean structure $\boldsymbol{x}_0$. Standard sampling processes will result in outputs distributed approximately as $p_{\text{data}}$, while our

underlying goal is instead to generate outputs distributed in $p_\mathcal{C}$. Generating from $p_\mathcal{C}$, requires the sampling procedure to be modified to incorporate constraints, as discussed in the next section.

## 3 CHALLENGES WITH CONSTRAINED DIFFUSION

As previously noted, diffusion models natively learn to reconstruct samples from an unconstrained data distribution $p_{\text{data}}$, which is misaligned with the true goal for constrained generation described by Equation ($\star$). In other domains, this gap is often addressed by formulating constrained sampling as an optimization problem, and recent work has extended this perspective to diffusion models by proposing a general framework for inference-time constrained generation (Christopher et al., 2024). The approach can be viewed as a sequential optimization problem:

$$\min_{\{x_t\}_{t=0}^T} \sum_{t=0}^T \ell_t(x_t, x_0) \quad \text{s.t. } x_t \in \mathcal{C} \ \forall t \tag{1}$$

where the single stage cost $\ell_t(x_t, x_0) := -\log p(x_t \mid x_0)$, and the constraint set $\mathcal{C}$ captures geometric or structural feasibility. Enforcing these constraints typically involves applying a projection operator $\Pi_\mathcal{C}(x_t) = \arg\min_{y \in \mathcal{C}} \|y - x_t\|_2^2$ after each reverse diffusion step. This is appealing because it embeds feasibility directly into the generative process; however, enforcing strict projections at every step raises two domain-specific challenges:

1. As observed by Blanke et al. (2025), projections on intermediate states introduces statistical biases. This arises as intermediate samples concentrate near constraint boundaries. This issue has also been reported for soft guidance schemes, where increased weight on the guidance terms tends to disrupt the diffusion trajectory and degrading performance (Dhariwal & Nichol, 2021; Nichol & Dhariwal, 2021; Ho & Salimans, 2022).
2. Additionally, intermediate feasibility requires projecting highly noisy states onto complex, non-convex constraints, which can result in solutions trapped in local minima (Pardalos & Vavasis, 1991; Boyd & Vandenberghe, 2004), disrupting the diffusion performance while simultaneously limiting the efficacy of feasibility updates.

Imposing constraints precisely on noisy samples may be reasonable under convexity assumptions, where the theoretical provisions apply, but within protein generation settings characterized by non-convex constraints, it is necessary to develop inference-time strategies which do not fundamentally rely on $x_t \in \mathcal{C}$. In the next section, we present our proposed method, designed intentionally with this principle in mind.

## 4 REVERSE DIFFUSION AS PROXIMAL OPTIMIZATION

To effectively sample constraint compliant outputs $x_0 \sim p_{\text{data}}(x_0) \mathbb{1}\{x_0 \in \mathcal{C}\}$ (Equation ($\star$)), we design an inference-time method that converges to $\mathcal{C}$, where the terminal distribution $\pi_0$ minimizes $\text{KL}(\pi_0 \mid p_\mathcal{C})$. Under this framing, a single reverse step of a diffusion sampler is viewed as an optimization problem in which the denoiser provides a data-driven "anchor", and feasibility is enforced by penalizing the distance to the constraint set.

Since $p_{\text{data}}$ is only available through a denoiser, the reverse process is realized incrementally. Each state $t$ of the reverse diffusion is composed of three stages: **(1) predict** the clean structure $\hat{x}_0^t$ from the current noisy state $x_t$, **(2) correct** this prediction by a proximal operator $\text{prox}_{\eta_t, g}$ that enforces feasibility, and **(3) renoise** the corrected clean structure with the forward kernel, denoted $\text{FWD}(\cdot, \varepsilon)$, to obtain the next noisy sample. The procedure is outlined in Figure 1 and summarized as:

$$x_t \xrightarrow{\text{predict}} \hat{x}_0^t \xrightarrow{\text{prox}} \tilde{x}_0^t \xrightarrow{\text{FWD}} x_{t-1}.$$

This modus operandi has strong theoretical properties as discussed in Section 6. First, a description of each step is detailed.

**1. Clean state prediction.** In protein design applications, it is common to employ an $x_0$-prediction parameterization. This design is intentional, as it enables adaptation from pretrained folding models like RoseTTFold, reusing their architectures and learned weights for initialization (Baek et al., 2021). At reverse time $t \in T, \ldots, 0$ the model takes a noisy latent $x_t$ and predicts a clean structure,

$$\mathbf{x}_\theta(x_t, t) = \hat{x}_0^t$$

providing an approximation of the final state. As $t \to 0$, predictions improve in accuracy as noise signal reduces. The availability of the predictor $\mathbf{x}_\theta$ is convenient as its output can be leveraged in our next step to restore feasibility.

**2. Feasibility step (proximal projection).** Next, feasibility requirements are applied on the predicted clean state. A feasible estimate is produced through a proximal map:

$$\tilde{\boldsymbol{x}}_0^t = \text{prox}_{\eta_t, g}(\hat{\boldsymbol{x}}_0^t) := \arg\min_{\boldsymbol{x}} \frac{1}{2\eta_t} \|\boldsymbol{x} - \hat{\boldsymbol{x}}_0^t\|^2 + g(\boldsymbol{x}) \tag{2}$$

where $\eta_t > 0$ is a step size determined by the degree of trust in the denoiser's prediction at step $t$ and $g : \mathbb{R}^{3 \times d} \to \mathbb{R}^1$ is a feasibility potential.

A natural first choice would be to take $g$ as the indicator function of the feasible set. However, this would require exact projections onto a potentially nonconvex set, which can be ill-defined when $\hat{\boldsymbol{x}}_0^t$ lies far from $\mathcal{C}$. To avoid instability, the hard indicator is replaced by its Moreau envelope (Boyd & Vandenberghe, 2004), yielding the smooth penalty

$$g(\boldsymbol{x}) = \frac{\lambda_t}{2} \text{dist}_{\mathcal{C}}(\boldsymbol{x})^2 \quad \text{with } \lambda_t > 0 \tag{3}$$

where $\text{dist}_{\mathcal{C}}$ is a distance metric from the feasible set (e.g., in SE(3), it could be $\inf_{\boldsymbol{y} \in \mathcal{C}} \|\boldsymbol{x} - \boldsymbol{y}\|$). The parameter $\lambda_t > 0$ plays the role of an inverse smoothing radius: as $\lambda_t \to \infty$ the penalty enforces exact feasibility, and for finite $\lambda_t$ it softly biases toward $\mathcal{C}$.

**3. Forward renoising.** Having obtained a feasible estimate, the next step reintroduces noise by sampling from the forward marginal at $t-1$ conditioned on the corrected clean sample $\tilde{\boldsymbol{x}}_0^t$:

$$\boldsymbol{x}_{t-1} = \text{FWD}(\tilde{\boldsymbol{x}}_0^t, \varepsilon) = \sqrt{\bar{\alpha}_{t-1}} \tilde{\boldsymbol{x}}_0^t + \sigma_{t-1} \varepsilon \tag{4}$$

with $\varepsilon \sim \mathcal{N}(0, I)$. This guarantees that, conditioned on $\tilde{\boldsymbol{x}}_0^t$, the marginal of $\boldsymbol{x}_{t-1}$ matches the forward diffusion at time step $t-1$. Note that as $\sigma_t \to 0$ and $\bar{\alpha}_t \to 1$, the Markov chain terminates at a clean $\boldsymbol{x}_0 = \tilde{\boldsymbol{x}}_0^1$. If $g$ acts as in indicator function, exact feasibility is recovered, while otherwise $\boldsymbol{x}_0$ becomes arbitrarily close to $\mathcal{C}$ as $\lambda_t$ increases.

**Selecting the schedule.** It is instructive to connect the proximal subproblem (Equation (2)) to probabilistic reasoning. By modeling the network's clean error at step $t$ as Gaussian with variance $\eta_t I$:

$$p(\boldsymbol{x}_0 \mid \boldsymbol{x}_t) \propto \exp\left(-\frac{1}{2\eta_t} \|\boldsymbol{x}_0 - \hat{\boldsymbol{x}}_0^t\|^2\right)$$

then interpreting the penalty $g$ as a soft prior of the form $\propto \exp\{-g(\boldsymbol{x}_0)\}$, Equation (2) computes the *per-step MAP estimate* $\tilde{\boldsymbol{x}}_0^t$ of the clean state. The subsequent renoising step, Equation (4), reinstantiates the correct stochasticity for the reverse chain while anchoring it to constraint set $\mathcal{C}$.

Because $\sigma_t^2$ shrinks over time, it is natural to schedule $\lambda_t$ to grow, so that feasibility becomes dominant only when the model's $\hat{\boldsymbol{x}}_0^t$ is accurate. Similarly, if $\eta_t = \sigma_{t-1}^2$ the trust weight can be directly connected to the diffusion variance, and the clean proximal problem remains on the same scale.

This predict-prox-renoise step is the stochastic analogue of a proximal gradient step. As elaborated in Section 6, the predict-prox-renoise cycle both respects the diffusion dynamics and guarantees convergence to feasible terminal states.

## 5 DECOUPLING GLOBAL TOPOLOGY FROM LOCAL GEOMETRY VIA ADMM

The constraint set $\mathcal{C}$ captures a strong coupling between *local* stereochemical variables and *global* variables governing topology and long-range residue interactions. Because residues that are far apart in sequence may lie adjacent in the folded structure, enforcing global constraints thus necessitates coordinated updates that can significantly impact nearby stereochemistry. For instance, we observe that applying non-covalent bond constraints on a $\beta$-strand can cause the associated residues to shift substantially, degrading the fidelity of this local geometry (Budyak et al., 2024). These interdependencies make the proximal step computationally complex. However, the presence of these separable local and global constraints confers structure to the problem and thus presents an opportunity to exploit it, enabling the use of decomposition approaches.

Consider that a feasible point can be equivalently represented as $\boldsymbol{x} \in \mathcal{C}_{\text{local}} \cap \mathcal{C}_{\text{global}}$. The local constraints $\mathcal{C}_{\text{local}}$ capture properties that are applicable in all backbone design tasks (e.g., adherence to stereochemical bond lengths and angles between consecutive atoms and residues). The global constraints $\mathcal{C}_{\text{global}}$ are problem-specific functionals: for example, in our first experiment, this constraint set defines bond lengths and angles between specific non-neighboring residues, characterizing non-covalent bonds which are necessary for protein-ligand pocket design.

Following this intuition, the feasibility potential is decomposed as $g(\boldsymbol{x}) = g_{\text{local}}(\boldsymbol{x}) + g_{\text{global}}(\boldsymbol{x})$, where

$$g_{\text{local}}(\boldsymbol{x}) = \frac{\lambda_t}{2}\text{dist}_{\mathcal{C}_{\text{local}}}(\boldsymbol{x})^2, \qquad g_{\text{global}}(\boldsymbol{x}) = \frac{\lambda_t}{2}\text{dist}_{\mathcal{C}_{\text{global}}}(\boldsymbol{x})^2$$

and $g_{\text{local}}, \ g_{\text{global}} : \mathbb{R}^n \to \mathbb{R} \cup \{+\infty\}$. These functions are proximable; squared distance penalties are adopted, which are treated as indicator functions when $\lambda_t \to \infty$. Hence, our proximal update is reframed as:

$$\Pi(\hat{\boldsymbol{x}}_0^t) := \arg\min_{\boldsymbol{x}} \underbrace{\frac{1}{2\eta_t}\|\boldsymbol{x} - \hat{\boldsymbol{x}}_0^t\|^2 + g_{\text{local}}(\boldsymbol{x})}_{=:F(\boldsymbol{x})} + \underbrace{g_{\text{global}}(\boldsymbol{x})}_{:=G(\boldsymbol{x})}.$$

Crucially, we *define* the local block $F$ to include the distance-to-denoiser term so that the local step both repairs stereochemistry and stays close to $\hat{\boldsymbol{x}}_0^t$, while the global block $G$ focuses on long-range feasibility. We solve this by a consensus ADMM on

$$\min_{\boldsymbol{y}, \boldsymbol{z}} \ F(\boldsymbol{y}) + G(\boldsymbol{z}) \quad \text{s.t.} \quad \boldsymbol{y} = \boldsymbol{z}, \tag{5}$$

with scaled dual variable $\boldsymbol{u}$ and penalty $\rho > 0$. This leads to the proximal splitting form of ADMM (Douglas–Rachford) (Parikh et al., 2014), with the update

$$\boldsymbol{y}^{k+1} := \text{prox}_{\rho^k, F}(\boldsymbol{y}^k - \boldsymbol{u}^k), \tag{6a}$$

$$\boldsymbol{z}^{k+1} := \text{prox}_{\rho^k, G}(\boldsymbol{z}^k + \boldsymbol{u}^k), \tag{6b}$$

$$\boldsymbol{u}^{k+1} := \boldsymbol{u}^k + \boldsymbol{y}^{k+1} - \boldsymbol{z}^{k+1} \tag{6c}$$

where $k$ is an iteration counter and $\boldsymbol{y}^0 = \boldsymbol{z}^0 = \hat{\boldsymbol{x}}_0^t \in \mathbb{R}^{3 \times d}$. Here $\boldsymbol{y}$ and $\boldsymbol{z}$ are two copies of the backbone, associated with $F$ and $G$ respectively, and the dual variable $\boldsymbol{u}$ accumulates their mismatch. At convergence the iterates satisfy $\boldsymbol{y} = \boldsymbol{z}$, recovering the minimizer of $F + G$; in practice it is only necessary to take a single sweep per diffusion step, but warm-starting across steps ensures the two copies remain close.

These updates can, thus, be interpreted as applying ADMM to the consensus problem, where the dual variable $\boldsymbol{u}$ carries forward residuals between local and global feasibility corrections. In practice, this is implemented by minimizing the associated augmented Lagrangian. For clarity, we present only the proximal form here but detail the explicit augmentented Lagrangian in Appendix G.

## 6 THEORETICAL ANALYSIS

Now, we show that the samples generated by the proposed stochastic proximal method come with feasibility guarantees. In the following we assume that the constraint set $\mathcal{C}$ is prox-regular and defer all proofs in Appendix H.

We start by providing a bound on the feasibility guarantees attained by the generated final sample.

**Theorem 6.1.** *Consider a feasibility potential* $g(\boldsymbol{x}) = \frac{\lambda_t}{2}\text{dist}_{\mathcal{C}}(\boldsymbol{x})^2$ *defined as in Equation 2. Then, the proximal minimizer* $\tilde{\boldsymbol{x}}_0$ *satisfies:*

$$\underbrace{\text{dist}_{\mathcal{C}}(\tilde{\boldsymbol{x}}_0)}_{\text{feasibility}} \leq \tfrac{1}{\sqrt{2\lambda_t \eta_t}}\text{dist}_{\mathcal{C}}(\hat{\boldsymbol{x}}_0) \tag{7}$$

The inequality shows that the proximal step contracts the violation by $(2\lambda_t \eta_t)^{-1/2}$, guiding the reverse process towards the constraint set. Then, as $t \to 0$ and $\lambda_t \eta_t \to \infty$, the corrected iterate converges arbitrarily close to the constraint set.

The following result provides rationale for how to schedule $\lambda_t$.

**Theorem 6.2.** *Let $K$ be a finite number such that $\mathbb{E}\big[\mathrm{dist}_{\mathcal{C}}(\hat{\boldsymbol{x}}_t, t)^2\big] \leq K$ for all $t$, then choosing $\lambda_t = \frac{c_t}{\eta_t}$ with a non decreasing $c_t$ yields:*

$$\mathbb{E}\big[\mathrm{dist}_{\mathcal{C}}(\tilde{\boldsymbol{x}}_0^t)^2\big] \leq \frac{K}{2c_t} \quad \text{and hence} \quad \mathbb{E}\big[\mathrm{dist}_{\mathcal{C}}(\tilde{\boldsymbol{x}}_0)^2\big] \leq \frac{K}{2c_1} \tag{8}$$

As a consequence, tightening $c_t$ towards the end guarantees decreasing expectation over the violations, leading to arbitrary small terminal violations. Note also that taking $g_1$ as the identity function over the constraint set $C$ gives *exact feasibility*.

Beyond these quantitative feasibility bounds, the per-step proximal subproblem needs to be well-posed. In particular, the question becomes whether a (possibly local) minimizer of the proximal mapping exists under the modeling assumptions.

**Theorem 6.3.** *Consider the proximal subproblem*

$$\mathrm{prox}_{\eta_t, g}(\boldsymbol{x}) = \tfrac{1}{2\eta_t}\|\boldsymbol{x} - \hat{\boldsymbol{x}}_0^t\|^2 + \tfrac{\lambda_t}{2}\,\mathrm{dist}_{\mathcal{C}}(\boldsymbol{x})^2,$$

*with $\eta_t, \lambda_t > 0$ and $\mathcal{C} \subset \mathbb{R}^{3\times n}$ nonempty and closed. Then:*

1. ***Existence.*** $\mathrm{prox}_{\eta_t, g}$ *is continuous and coercive, hence attains a global minimizer for every $\hat{\boldsymbol{x}}_0^t$. In particular,* $\arg\min \mathrm{prox}_{\eta_t, g} \neq \varnothing$.
2. ***Local uniqueness.*** *If $\hat{\boldsymbol{x}}_0^t$ lies within the prox-regularity neighborhood of $\mathcal{C}$, then the projection $\Pi_{\mathcal{C}}(\hat{\boldsymbol{x}}_0^t)$ is single-valued. Moreover, if $\mathrm{prox}_{\eta_t, g}$ is strongly convex in a neighborhood of $\Pi_{\mathcal{C}}(\hat{\boldsymbol{x}}_0^t)$, then the proximal minimizer is unique within that neighborhood.*

The analysis shows that feasibility improves monotonically as $\lambda_t$ tightens relative to $\eta_t$, while the proximal subproblem remains well-posed due to the quadratic distance term. In practice, this provides principled guidelines for selecting schedules which balance denoiser trust against constraint enforcement. The next section demonstrates that the theoretical foundations translate to tangible performance improvements in protein backbone design.

# 7 EXPERIMENTS

We conduct our evaluation on two key tasks for protein motif scaffolding in the PDZ domain and vacancy-constrained pocket design for unconditional generations, described in more details in Sections 7.1 and 7.2.

**Baselines.** The comparison includes state-of-the-art structure-based design methods incorporating constraint conditioning. We use *RFDiffusion* (Watson et al., 2023) as the underlying backbone across all baselines and our method, although we note that any method evaluated could be extended to other backbone structure diffusion models (Yim et al., 2023; Cutting et al., 2025). RFDiffusion is selected, as it's considered a state-of-the-art approach for the tasks considered in this work:

1. **Standard Diffusion:** RFDiffusion conditioned on relevant motifs or structures, as it is often used in practice.
2. **Recentering of Mass Guidance:** RFDiffusion with conditioning to bias the generation towards a particular Cartesian coordinate where bond interactions or cavity-defining residues enforce constraint satisfying formations (Braun et al., 2024). Conditioning is fine-tuned to find the lowest average constraint violation per sample prior to beginning experiments (see Appendix B).
3. **Constraint-Guided Diffusion (CGD):** RFDiffusion with constraint-based guidance Sequential Monte Carlo sampling (Lee et al., 2025). This SOTA method uses importance sampling with a guided rate matrix to define weights and periodic resampling based on constraint violations.

**Metrics.** The following metrics are adopted to evaluate the performance of our method:

1. **Constraint Satisfaction:** The percentage of samples satisfying all domain-specific constraints. For Section 7.1, we verify presence of the described hydrogen-bond pairing of the backbone through DSSP (Kabsch & Sander, 1983), while Section 7.2 computes constraints via Cartesian coordinate bounds.
2. **Structure Realism:** Percentage of samples containing secondary structures within constrained regions while maintaining general backbone realism (e.g., $\beta$-sheets remain less than ten residues, inter-residue distances and angles are preserved).

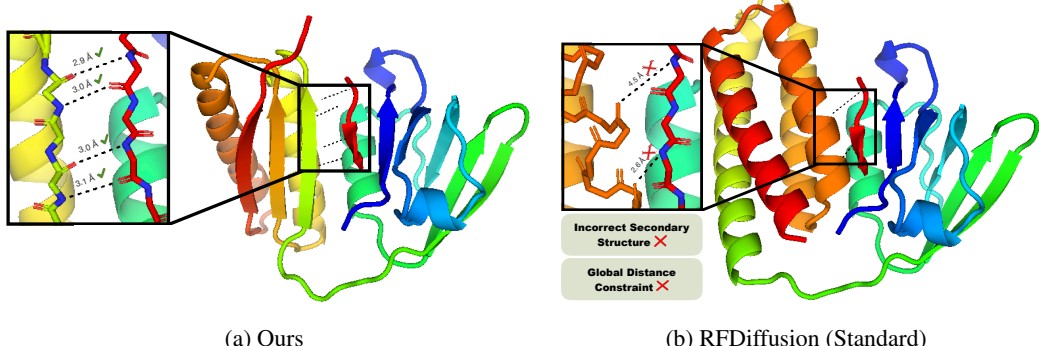

| (a) Ours | (b) RFDiffusion (Standard) |

Figure 2: Visualization of randomly selected samples generated by (a) our proximal method and by (b) Standard RFDiffusion on our introduced PDZ domain benchmark.

3. **Usable Percentage:** Percentage of samples passing above conditions for Structure Realism and Constraint Violation. Indicates frequency of generating a structure that satisfies physical plausibility and functional requirements.

4. **Radius of Gyration:** Average radius of gyration across generated backbones, measuring the overall spatial compactness of the structure. Lower values typically correspond to more compact, globular folds, while higher values indicate extended or unfolded conformations.

5. **Diversity:** Percentage of samples which are both useable and satisfy a minimum root mean squared error between all other samples (2 Å). Higher diversity indicates better coverage of possible structure.

Additionally, we note that further details on all experimental setups are provided in Appendix B.

## 7.1 PDZ Domain: Non-covalent Bonds

Antibodies are widely used in molecular biology to target specific proteins, but their large size and extracellular restriction limit their utility, particularly in intracellular contexts. In addition, their binding interfaces, often mediated by flexible loop regions, pose challenges for computational design. As an alternative, biology employs small modular protein domains (e.g., WW, SH2, SH3, PH, and PDZ), which provide more designable binding modes. PDZ domains, for instance, recognize unstructured C-terminal motifs of partner proteins, typically through $\beta$-sheet–like hydrogen bond contacts. These interactions are generally weak and promiscuous, serving primarily in protein localization. To enhance affinity and specificity, Huang et al. (2009) engineered concatenated PDZ fusions with other small domains, demonstrating improved performance. Protein diffusion models extend this concept by enabling conditioning on both PDZ domains and their peptide ligands, thereby facilitating the *de novo* design of concatenated architectures.

**Dataset construction.** To benchmark constrained diffusion for PDZ engineering, we collected all resolved PDZ/PDZ binding motif (PBM) complexes from the RCSB Protein Data Bank (Berman et al., 2000). Seventy-two structures were initially retrieved and manually curated to remove entries with unresolved regions or peptides too short for recognition, yielding fifty-two usable complexes. To expand the existing PDZ domain in a reliable way to make additional contacts with the target PBM, the N- and C-termini had to be rearranged. Each structure was processed to reposition termini closer to the bound peptide by introducing a cut in a loop adjacent to the ligand, trimming the original termini, and applying vanilla RFDiffusion to in-paint the resulting gap. Candidate backbones were filtered to exclude chain breaks and non–$\beta$-sheet pairings. Sequences for the redesigned regions were generated using ProteinMPNN, followed by structure prediction with AlphaFold2. Predicted models were retained only if they satisfied stringent criteria: (i) self-consistency RMSD to the RFDiffusion backbone <2.5 Å, (ii) mean pLDDT >90, and (iii) peptide RMSD <2.0 Å. After filtering, 31 high-confidence PDZ designs remained for benchmarking. While all 31 structures are valid targets for this approach, six of them (PDB IDs: 2AWW, 2G2L, 4JOG, 6UBH, 6Y9O, 8CN3) contain short peptide ligands or peptides with geometrically restrictive residues such as prolines. These structures remain in the benchmark set, but they are distinguished in the results as having poor-posed ligands. More details are provided in Appendix C. This benchmark for constrained diffusion is also a novel contribution of this work.

| | RFDiffusion | | | Ours |
|---|---|---|---|---|
| | Standard | Recenter | CGD | |
| Constraint Satisfaction (%) [↑] | 0.0 | 0.0 | 0.0 | **100.0** |
| Structure Realism (%) [↑] | (32.0) | (18.7) | (38.2) | **21.0** |
| Usable Percentage (%) [↑] | 0.0 | 0.0 | 0.0 | **21.0** |
| Radius of Gyration (Å) [↓] | (13.6) | (13.2) | (16.2) | **12.4** |
| Diversity (%) [↑] | N/A | N/A | N/A | **18.8** |

Table 1: Comparison to structure-based design baselines, Standard (Watson et al., 2023), Recenter (Braun et al., 2024), and CGD (Lee et al., 2025), and ours, for the *PDZ domain*. Results reported across 31,000 samples for each baseline, highlighting **best** and second best results. Parentheses indicate when statistics are computed over **unusable** structures.

**Task description.** Given a target PDZ domain and its peptide ligand, the objective is to design an additional domain concatenated to the PDZ that contributes new stabilizing contacts with the peptide. This requires satisfying global inter-chain constraints. Specifically, because the peptide forms $\beta$-sheet-like contacts with the PDZ domain, we aim to create a complementary set of $\beta$-sheet-like interactions on the opposite face of the peptide using the engineered concatenated domain. Constraints are enforced to ensure valid bond lengths and angles, while also ensuring local stereochemisty is preserved. Success is evaluated by the metrics aforementioned, where global feasibility is defined by ideal bond lengths ($2.9 \pm 0.2$ Å), C=O$\cdots$N angles ($155 \pm 10°$), and C$_\alpha$-N$\cdots$O ($120 \pm 10°$).

**Results.** Table 1 provides results on the PDZ benchmark comparing our constrained diffusion approach (visualized in Figure 2 (a)) to the baseline methods. Notably, across nearly one hundred thousand samples generated for the three baselines, not one sample perfectly satisfied the bonding distance and angle constraints. The baselines frequently generate incorrect secondary structures, as illustrated in Figure 2 (b), making it implausible that generations will effectively bind with the peptide ligand. While recentering and CGD perform well in terms of the local geometric requirements captured by the structure realism measurement, they are unable to cope with the global requirement of non-covalent bonding between the PDZ backbone continuation and the peptide ligand, as reflect by the constraint satisfaction rates. While constraint-guided diffusion satisfies the bond distance constraints for some generations, it is never able to generate residues which appropriately meet the angle requirements. Importantly, the *baselines yield no usable generations* for any of the 31 structures in the benchmark. In contrast, our method achieves state-of-the-art results, generating usable structures in **21.0%** of total generations (and up to **83.0%** for well-posed ligands), markedly outperforming the existing baselines. In addition to observing *perfect constraint satisfaction, we outperform all methods substantially in radius of gyration and diversity metrics*. These results highlight the unique ability of our approach to handle both local stereochemical properties while enforcing global functional constraints, providing a vastly more viable approach to protein engineering under specific property requirements and design constraints.

## 7.2 MOLECULE ENCAPSULATION: VACANCY CONSTRAINTS

Several recent approaches have improved the usefulness of diffusion-based protein generation with the integration of all-atom models and catalytic site scaffolding (Krishna et al., 2024; Ahern et al., 2025; Braun et al., 2024). A key remaining limitation is precise spatial control over where new structure is placed. Adjusting the diffusion origin can help, but explicit user control over inclu-

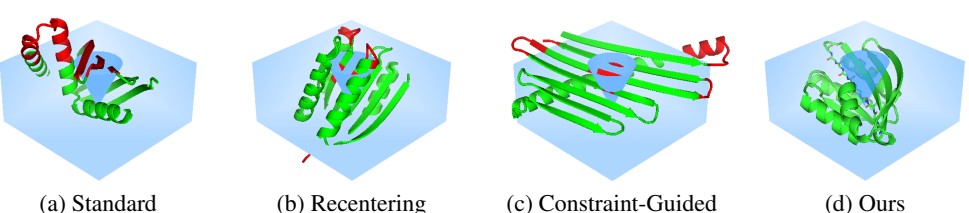

| (a) Standard | (b) Recentering | (c) Constraint-Guided | (d) Ours |
|---|---|---|---|

Figure 3: Visualization of randomly selected samples for the molecule encapsulation experiment; green parts of the structure fall within feasible regions, while the red parts violate the constraints.

| | RFDiffusion | | | | Ours |
|---|---|---|---|---|---|
| | Standard | Recenter | CGD | Recenter + CGD | |
| Constraint Satisfaction (%) [↑] | 0.0 | 0.0 | 21.6 | 27.4 | **100.0** |
| Structure Realism (%) [↑] | (100.0) | (100.0) | 96.1 | 93.8 | **97.8** |
| Usable Percentage (%) [↑] | 0.0 | 0.0 | 20.5 | 24.2 | **97.8** |
| Radius of Gyration (Å) [↓] | (15.2) | (14.3) | 23.9 | 26.6 | **14.8** |
| Diversity (%) [↑] | N/A | N/A | 20.5 | 24.2 | **97.8** |

Table 2: Comparison to structure-based design baselines: Standard (Watson et al., 2023), Recenter (Braun et al., 2024), and CGD (Lee et al., 2025), and ours, for *molecule encapsulation*. Results reported across 4000 samples, highlighting **best** and second best results. Parentheses indicate when statistics are computed over **unusable** structures.

sion/exclusion volumes would better enable tasks such as shaping small-molecule pockets, peptide-binding grooves, or membrane-embedded features. For example, Braun et al. (2024) approximated pocket formation by inserting a placeholder $\alpha$-helix to occupy volume during generation and deleting it afterward. Generalizing this idea to geometric volume constraints, hard inclusion masks and forbidden regions, could provide finer control and higher success rates in targeted protein design.

**Task description.** Given a fixed spatial environment defined by a rectangular box with an internal conical exclusion zone, the goal is to design protein backbones which fall exclusively in this nonconvex region while preserving the local geometries and secondary structures. Feasible structures are characterized by all atoms falling within the defined box ($20\,\text{Å} \times 40\,\text{Å} \times 40\,\text{Å}$), while simultaneously avoiding the exclusion zone introduced by the displacement (visualized in Figure 3).

**Results.** Figure 3 provides a visualization of representative samples from each baseline. Standard diffusion and recentering of mass guidance perform similarly in this domain, yielding realistic structures but failing to satisfy the functional requirements in any out of 1000 samples generated for each. We observe slightly different failure modes: standard diffusion generally violates the box constraint, while recentering more often violates the vacancy constraint. The recentering is effective at keeping the structures inside the box, but it cannot capture the exclusion zone. Table 2 reports much stronger performance for constraint-guided diffusion, especially when augmented with recentering, which generates feasible samples 27.4% of the time, resulting in 24.2% usable samples. However, it is worth noting that the qualitative performance suffers, as the radius of gyration is much higher than other baselines. This is often indicative of structures which contain unfolded conformations, ultimately undermining structural stability and realism. In comparison, our method reports *perfect constraint satisfaction*, producing $4\times$ **as many usable samples** as the nearest baseline, with an impressive **97.8%** success rate. Furthermore, it maintains radii of gyration comparable to standard diffusion, indicating the generated samples combine structural plausibility, compactness, and fold coverage.

# 8 RELATED WORK

While existing *de novo* protein structure design models produce plausible generations, as we have shown in this paper, sampled protein backbones frequently violate inter-atomic bond lengths, angles, or chain closure requirements, often necessitating the generation of tens of thousands of candidates to obtain a handful of viable designs (e.g., Watson et al. 2023; Sappington et al. 2024). Although backbone generators such as RFDiffusion (Watson et al., 2023; Ahern et al., 2025) provide major advances in functional conditioning, outputs still require post hoc filtering to ensure stereochemical correctness, and current pipelines continue to rely heavily on rejection sampling. Recent models such as Genie 2 and OriginFlow further illustrate the growing interest in diffusion and flow-based approaches for backbone generation (Lin et al., 2024; Yan et al., 2025); however, these methods exhibit similar limitations, with generated backbones often violating global constraints and requiring substantial downstream filtering.

Training-time methods have been proposed to address these issues by embedding structural constraints into generative models (Eguchi et al., 2022; Lutz et al., 2023). However, because protein design requires task-specific constraints, a model trained on one constraint set does not generalize, making broad applicability impractical without retraining. Furthermore, training-time approaches typically provide only distributional guarantees, biasing samples on average rather than ensuring per-sample feasibility. Models such as ReQFlow (Yue et al., 2025) and FoldFlow-2 (Huguet et al.,

2024) provide valuable tools, but likewise do not directly enforce hard geometric constraints, instead incorporating them as soft biases.

Inference-time approaches have emerged as a strong alternative, enforcing per-sample compliance and removing the need for model retraining. Diffusion guidance methods were first introduced for soft constraint imposition but are fundamentally limited, offering only probabilistic bias rather than guaranteed adherence to the constraint set (Ho & Salimans, 2022). While these techniques have improved performance for protein backbone generation, with models such as Chroma (Ingraham et al., 2023) leveraging conditioning on specific substructures, as we have shown in our experiments, they are often ineffective in providing consistent constraint satisfaction. Overcoming these limitations requires generative models which can effectively integrate these constraints into the design process, as presented in this work.

## 9 CONCLUSION

Motivated by the significant challenge of integrating functional design constraints into protein engineering tasks, this paper present a constrained diffusion framework for structure-guided design. By applying proximal feasibility updates with ADMM decomposition, the approach couples local stereochemical property enforcement with global utility requirements. To assess the quality of existing solutions as compared to the methodology presented in this paper, the work introduces a novel curated benchmark for protein motif scaffolding in PDZ domains, providing the first standard for constrained diffusion methods in modular domain engineering. Evaluation reports state-of-the-art results across motif scaffolding and vacancy-constrained pocket design, illustrating the ability of this approach to generate high quality proteins which adhere to precise domain-centric constraints.

## ETHICS STATEMENT

This work develops methods for constrained generative modeling in protein design. Our method improves feasibility in backbone design tasks, which holds significant potential to accelerate existing protein engineering pipelines. To mitigate risks of potential misuse, this paper restricts evaluation to safe, publicly available structural benchmarks, curated from the Protein Data Bank. All implementations follow standard open science practices, ensuring safe and transparent research

## REPRODUCIBILITY STATEMENT

The code and benchmark are released in the supplementary material along with instructions to guide the reproduction of the results presented in this paper. Methodological details are described extensively in the paper and accompanying appendix. Section 7, Appendix B, Appendix F describe the specific constraints, hyperparameters, models, and hardware used for evaluation pipelines.

## ACKNOWLEDGMENTS

This research is partially supported by NSF awards 2533631, 2401285, 2334936, and 2226816, by DARPA under Contract No. #HR0011252E005, by the National Institutes of Health, National Institute of General Medical Sciences, under Award Number T32 GM135141, and Sargassum BioRefinery (SaBRe) Center, a project of Schmidt Sciences' Virtual Institute of Feedstocks of the Future (VIFF), with support from the Foundation for Food & Agriculture Research. The authors acknowledge the Research Computing at the University of Virginia. Any opinions, findings, conclusions, or recommendations expressed in this material are those of the authors and do not necessarily reflect the views of NSF, DARPA, or the National Institutes of Health.

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

## A   LIMITATIONS AND FUTURE WORK

In scientific domains such as protein engineering, exact constraint satisfaction is essential for both physical realism (local constraints) and functional utility (global constraints). However, it is valuable to acknowledge several important trade-offs associated with the adoption of constrained methods. First, enforcing feasibility inherently introduces increased computational complexity. Sampling from $p_{\mathcal{C}}$ is a fundamentally harder task than sampling from $p_{\text{data}}$, especially under the assumption of hard constraint satisfaction, as reflected in reported runtimes (Appendix F). While runtime is worth considering, especially as the method scales, it is important to note that when sampling from $p_{\mathcal{C}}$, our approach offers the most efficient runtime, providing much better performance than the current rejection sampling paradigm. Although not applicable in our experimental settings, if sampling from $p_{\text{data}}$ is sufficient, it is not computationally tractable to adopt constrained diffusion methods.

As second caveat concerns theoretical assumptions. Our analysis assumes prox-regularity of the constraint set to provide formal guarantees. This condition is standard in proximal theory, but it may not apply for highly nonconvex constraints encountered in protein design tasks. In such cases, convergence guarantees do not explicitly apply, and solutions are instead justified empirically. However, the development of formal guarantees for nonconvex constraint sets remains an open and largely unexplored direction, as existing optimization theory cannot provide guarantees for general nonconvex sets.

Finally, while our experimental evaluation illustrates state-of-the-art performance on two highly nontrivial and practically significant protein engineering problems, it necessarily leaves many other design challenges open for future exploration. The problems we consider already capture central difficulties in enforcing exact feasibility under realistic structural and functional constraints, making them representative of some of the hardest settings encountered in practice. Extensions such as placing global constraints on higher-level properties (e.g., polarity of specific regions in the structure) or imposing specific formation constraints on secondary structures (e.g., controlling the radius of a $\beta$-barrel) are complimentary next steps. We view these as exciting opportunities for future work, with the present study establishing a critical foundation for handling such broader classes of design challenges.

## B   EXPERIMENTAL DETAILS

| Task | $\boldsymbol{u}^0$ | $c_T$ | $c_1$ | $T$ | Diffusion schedule |
|---|---|---|---|---|---|
| *PDZ Domain* | 0.0 | finite ($\approx$ tol. 3.0 Å) | $\infty$ | 45 | linear in $t$, from $\bar{\alpha}_0 \approx 1$ to $\bar{\alpha}_T \approx \prod_t (1 - 7e{-}2)$; SO(3): 1.5→2.5; $\sigma \in [0.02, 1.5]$ |
| *Molecule Encapsulation* | 0.0 | $\infty$ | $\infty$ | 50 | linear in $t$, from $\bar{\alpha}_0 \approx 1$ to $\bar{\alpha}_T \approx \prod_t (1 - 7e{-}2)$; SO(3): 1.5→2.5; $\sigma \in [0.02, 1.5]$ |

Table 3: Experiment hyperparameters.

All hyperparameters are reported in Table 3. Both tasks use RFDiffusion defaults: $T = 50$ steps, linear $\beta$ schedule from $10^{-2}$ to 0.07, SO(3) schedule 1.5–2.5, coordinate scaling 0.25, and Gaussian noise $\sigma \in [0.02, 1.5]$. Proximal multipliers $c_t$ tighten from $c_T$ (finite) to $c_1 = \infty$ across the trajectory. Other hyperparameters $\lambda$ and $\eta$ can be derived by Theorem 6.2.

### B.1   PDZ DOMAIN: NON-COVALENT BONDS

The evaluation is conducted across our newly introduced dataset, including 31 distinct PDZ structures. For each structure, we generate 1000 samples with each of the baselines, yielding a total of 31,000 designed proteins. From these generations, we assess the overall performance of each method, leveraging PyRosetta to conduct the final evaluation of the generations (Chaudhury et al., 2010).

**Recenter of Mass Guidance.**   Prior to running the evaluation, we search the local space surrounding the peptide ligand to empirically optimize the selected center of mass. We generate ten samples for each selected point in space, providing representative data for the performance on different conditioning. Selection is determined based on the lowest average constraint violation (capturing angles and bond lengths between the peptide ligand and the sampled generations). After completing this search, this center of mass is used for all runs on the particular motif.

**Constraint-Guided Diffusion.** Adopting the Sequential Monte Carlo conditioning proposed by Lee et al. (2025), we set the sample weighting via the constraint violation of the clean predicted state $\hat{x}_0^t$. For the evaluation, we fix the number of particles $P = 200$, as this seems to balance potential performance and overall runtime. We adopt a multinomial resampling function and introduce an inverse temperature parameter, $\beta$, which controls the sharpness of the resampling distribution. Candidate weights are computed as

$$w_i \propto \exp\left(-\beta\left(c_i - \min_j s_j\right)\right)$$

where $c_i$ denotes the score of constraint violation $i$. Earlier in the diffusion process, $\beta = 30$ remains high, keeping the resampling fairly stochastic, but we lower this $\beta = 1$ as $t \to 0$ to improve selection for the final state and increase the likelihood of non-covalent bonds forming.

**Constrained Diffusion (Ours).** As described in Section 5, our projection is implemented through an ADMM derived decomposition. We assign $y$ to be consecutive residues which fall between the start and end of the bonding points on the peptide ligand. For instance, if the peptide has six covalent bonding sites, $y$ is composed of five residues. In this case, O–N and N–O bonds are placed on the first, third, and fifth residues. Then, $z$ is composed of the linker preceding $y$ (typically six residues), and the chain continuation following $y$ (typically 60-100 residues). Additionally, we note that our approach leverages constraint guidance for this experiment, as our approach is complementary to these guidance methods.

### B.2 MOLECULE ENCAPSULATION: VACANCY CONSTRAINTS

For this setting, we evaluate across 1000 samples for each of the baselines. We use a fixed box of size 20 Å$\times$ 40 Å$\times$ 40 Å, centering the apex of the cone 5 Å above the bottom face of the box, with a half-angle of $25°$. As we identify that many *baseline* samples violate only the width constraints of the box (40 Å$\times$ 40 Å), which is not integral to the application, we do not report these violations in our evaluation provided in Figure 2.

**Recenter of Mass Guidance.** The center of mass is positioned at the center of the box, encouraging the sample to (1) remain inside of the box and (2) maximize contact with the cone shaped vacancy. Similar to the previous setting, the guidance is tuned on a small set of sample prior to generating outputs for evaluation; however, for this experiment the center of mass is fixed during tuning, only searching over the *guidance strength* parameter.

**Constraint-Guided Diffusion.** We adopt an identical setup for the constraint guidance as described for the PDZ domain, but leveraging an identical constraint violation measure:

$$g(x) = \inf_{y \in \mathcal{C}} \|x - y\|^2$$

which captures the sum of the distance of each atom from the feasible region. We similarly adopt a particle count of $P = 200$, and follow an identical schedule to the previous experiment for the inverse temperature.

**Constrained Diffusion (Ours).** In this setting, all atomic positions are subject to the global constraints, making $y = x$, removing the need for explicit variable splitting in this setting. While we note that the vacancy constraint is indeed nonconvex, the projection operator can be represented in closed form, as it is composed of simple convex shapes; this alleviates concerns surrounding the tractability of the projection operator, which is the primary motivation for decoupling global and local requirements. To preserve local geometric properties and secondary structures, we extend this operator by enforcing rigid-body consistency on secondary structure segments, while adjoining linker regions are adjusted to absorb residual displacements. This construction ensures that the projection both satisfies global constraints and maintains local stereochemical integrity. Additionally, we note that our approach leverages constraint guidance and recentering of mass for this experiment, as our approach is complementary to these guidance methods.

**Additional Results.** To supplement the results reported in the paper, we include additional analysis of the secondary structure preservation rates for samples generated by each baseline and by our method. This metric quantifies the percentage of residues assigned to a canonical secondary structure state in the relaxed models. Higher values indicate improved preservation of fold-level features.

| | RFDiffusion | | | Ours |
| --- | --- | --- | --- | --- |
| | **Standard** | **Recenter** | **CGD** | |
| Secondary Structure (%) | 80.8 | 82.5 | 83.5 | **85.0** |

Table 4: Secondary structure preservation across design methods.

In addition to the primary metrics reported in the main text, we assess the structural plausibility of generated designs using a secondary structure preservation metric. This measure is assessed using PyRosetta to determine the percentage of residues which fall in helix, sheet, or turn conformations in the relaxed structures (Chaudhury et al., 2010). As shown in Table 4, our method achieves the highest secondary structure frequency, suggesting that the model maintains global fold characteristics more reliably than the baselines.

## C  DATASET CONSTRUCTION

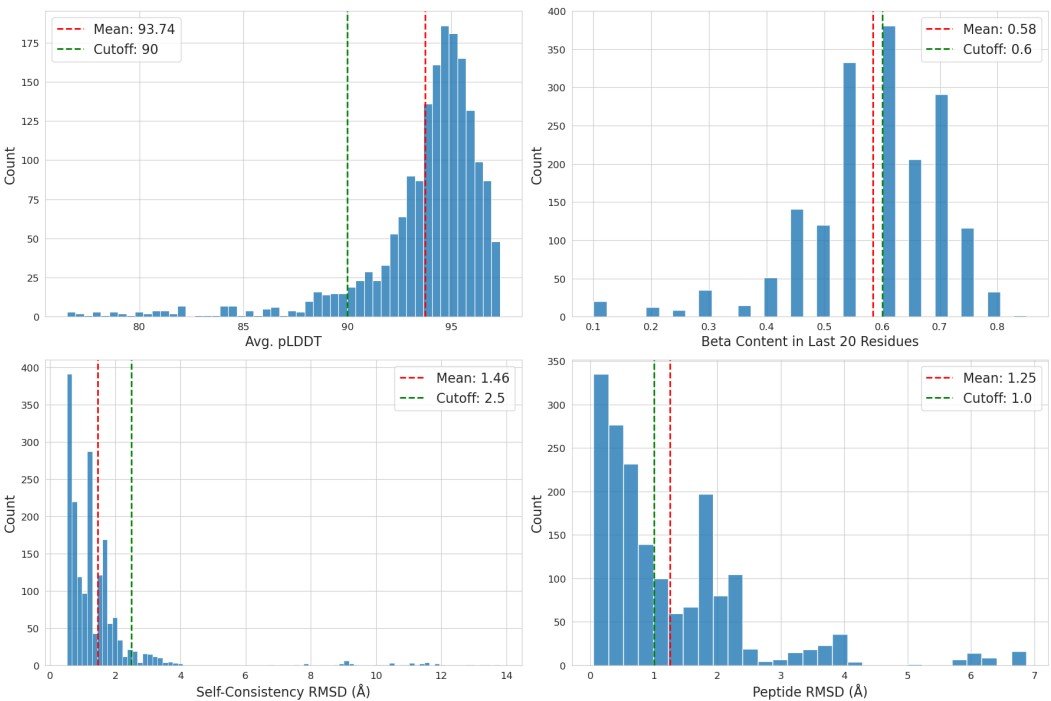

Figure 4: Dataset filtering details expanded.

Details for the filtering are included in Figure 4. For every candidate structure per target PDZ-PBM loop closure, AlphaFold models were constructed. To determine the confidence of each model, predicted Local Distance Difference Test (pLDDT) was collect. The cutoff for pLDDT for a viable loop closure was set to >90 (top left). The former N- and C- termini form a $\beta$-sheet pairing, to conserve this, a filter of >60% of the residues being $\beta$-sheet was used (top right). ProteinMPNN was used to generate potential sequences that conform to generated backbones for the in-painted region. After having AlphaFold predict the structure given the new sequence, RMSD was measured between the generated backbone and the predicted structure. A cutoff of <2.5Å was used (bottom left). To ensure the peptide was not interfered with after predicting the new structure with AlphaFold, a stringent cutoff of <1Å RMSD was used (bottom right)

## D   COMPARISON TO PROJECTED DIFFUSION MODELS

As discussed in Section 3, prior work on constrained generative sampling has proposed Projected Diffusion Models (Christopher et al., 2024). These models augment the sampling process by imposing a projection onto the feasible set after each diffusion step. While theoretical guarantees have been derived for convex constraint sets, and empirical observations have illustrated that these properties hold for certain nonconvex sets, the paper has argued critical limitations for step-then-project approaches when applied to challenging settings as explored in this work. This section presents empirical justification of these aforementioned claims.

| | Projected Diffusion Model | | Ours |
|---|---|---|---|
| | **Local + Global** | **Global** | |
| Constraint Satisfaction (%) | 0.0 (Failed to Converge) | **100.0** | **100.0** |
| Structure Realism (%) | N/A | 0.0 | **97.8** |
| Usable Percentage (%) | 0.0 | 0.0 | **97.8** |
| Radius of Gyration (Å) | N/A | (13.1) | **14.8** |
| Diversity (%) | N/A | N/A | **97.8** |

Table 5: Comparison to Projected Diffusion Models on the Molecule Encapsulation setting.

The results in Table 5 reaffirm the limitations of integrating intermediate projection steps when working with high-dimensional data subject to complex nonconvex constraints. While in this setting it was possible to exactly satisfy the global constraints, the resulting structures are unusable, with high concentrations of atoms placed on the edges of the box and cone boundaries. When also enforcing the local constraints explicitly, the projection operator never converged on any samples or timesteps.

## E   HYPERPARAMETER SENSITIVITY ANALYSIS

To assess the robustness of our method to hyperparameter choices, we performed two ablation studies: (1) varying constraint weight schedule $\lambda$, and (2) varying the number of ADMM iterations per diffusion step. All experiments were conducted on the PDZ domain design target 6ubh, a well-posed example from our benchmark set suited to constrained generation.

| | $\lambda = \frac{10}{t}$ | $\lambda = \frac{100}{t}$ | $\lambda = \frac{1000}{t}$ | $\lambda = \inf$ |
|---|---|---|---|---|
| Constraint Satisfaction (%) | **100.0** | **100.0** | **100.0** | **100.0** |
| Structure Realism (%) | 80.0 | **100.0** | **100.0** | 90.0 |
| Usable Percentage (%) | 80.0 | **100.0** | **100.0** | 90.0 |
| Radius of Gyration (Å) | 10.58 | 10.53 | 10.56 | **10.34** |
| Diversity (%) | 80.0 | **100.0** | **100.0** | 90.0 |

Table 6: Ablation over constraint-weight schedules $\lambda$ on PDZ domain sample 6ubh.

As illustrated by Table 6, increasing the initial weight of $\lambda_T$ yields improvements in structural realism and usable sample percentage. This trend aligns with the interpretation that stricter constraint enforcement at early timesteps guides the diffusion trajectory toward the feasible manifold, improving geometric quality. However, increasing this too strictly, results in over constraining early states, analogous to placing an overly large weight on the diffusion-guidance terms as discussed in Section 3.

The next analysis centers on determining the optimal number of ADMM iterations per diffuson step. As described in Section 5, a single ADMM sweep per diffusion step is sufficient in practice. In fact, the ablation suggests that multiple iterations degrade quality. The failure case that emerges over additional ADMM iterations is actually always tied to the formation of a $\beta$-sheet across the binding sites; additional iterations do not result in local stereochemical properties breaking, but the analysis suggests that they do lower the liklihood of secondary structure conformations. This is because a higher wieght is placed on the ADMM optimization than the diffusion updates, leading to

|  | ADMM = 1 | ADMM = 2 | ADMM = 3 |
|---|---|---|---|
| Constraint Satisfaction (%) | **100.0** | **100.0** | **100.0** |
| Structure Realism (%) | **80.0** | 70.0 | 40.7 |
| Usable Percentage (%) | **80.0** | 70.0 | 40.7 |
| Radius of Gyration (Å) | **10.58** | 10.72 | 11.24 |
| Diversity (%) | **80.0** | 70.0 | 40.7 |

Table 7: Ablation on the number of ADMM iterations per diffusion step.

feasible generations but not ones which fall closely on the data manifold. Furthermore, we note that additional ADMM steps also become impractical because adding these iterations scales the runtime nearly linearly.

## F    RUNTIME

Benchmarks were run on a single slice of an NVIDIA A100 Multi-Instance GPU (MIG) to efficiently parallelize sampling across available hardware. We posit that overall runtimes could be improved by scaling the available resources, but this is beyond the scope of this study; for instance, both constraint-guided diffusion and our approach could be significantly parallelized to increase efficiency, as they operate over up to $P = 200$ particles. Note that reducing $P$ will accelerate performance, providing a tunable trade-off between speed and performance. Hence, we expect that if resource constraints could be ignored, these methods could be accelerated by multiple orders of magnitude.

|  | Standard | Recenter | CGD | *Ours* |
|---|---|---|---|---|
| | | *PDZ Domain* | | |
| Time Per Usable Sample (min) | $\infty$ | $\infty$ | $\infty$ | **935.2** |
| | | *Molecule Encapsulation* | | |
| Time Per Usable Sample (min) | $\infty$ | $\infty$ | 142.4 | **98.4** |

Table 8: Reported runtimes on both settings. Analysis considers the average time per usable sample.

Our method provides the best runtimes *per usable sample*. In protein design experiments, this is a more important metric than raw runtime, as only usable samples are relevant when considering overall performance. Hence, under the assumption that functional requirements are necessary, which is specifically the applications this work targets, **the proposed method is indeed the most computationally efficient!**

## G    ADMM

This section expands the discussion of the ADMM formulation introduced in Section 5. Begin by recalling the consensus formulation of our proximal update:

$$\text{prox}_{F+G}(\boldsymbol{y}, \boldsymbol{z}) := \arg\min_{\boldsymbol{y}, \boldsymbol{z}} F(\boldsymbol{y}) + G(\boldsymbol{z}) \qquad s.t.\ \boldsymbol{y} = \boldsymbol{z}$$

where $F$ includes the data anchor term and local feasibility requirements, and $G$ encodes global feasibility.

$$F(\boldsymbol{y}) = \frac{1}{2\eta_t}\|\boldsymbol{y} - \hat{\boldsymbol{x}}_0^t\|^2 + \frac{\lambda_t}{2}\text{dist}_{\mathcal{C}_{\text{local}}}(\boldsymbol{y})^2$$

$$G(\boldsymbol{z}) = \frac{\lambda_t}{2}\text{dist}_{\mathcal{C}_{\text{global}}}(\boldsymbol{z})^2$$

Recall that the consensus initialization $\boldsymbol{y}^0 = \boldsymbol{z}^0 = \hat{\boldsymbol{x}}_0^t \in \mathbb{R}^{3 \times n}$, which aligns the start of the iterates with the denoiser's prediction.

Classical ADMM results (see Parikh et al. 2014) guarantee convergence to a minimizer of $F + G$ under convexity assumptions. In our case, convexity does not strictly hold, but in practice this algorithm is often effectively applied to nonconvex settings, achieving feasible solutions.

The formulation yields the augmented Lagrangian, introducing the dual variable $\boldsymbol{u}$ and a penalty parameter $\rho$:

$$\mathcal{L}_\rho(\boldsymbol{y}, \boldsymbol{z}, \boldsymbol{u}) = F(\boldsymbol{y}) + G(\boldsymbol{z}) + \boldsymbol{u}^\top(\boldsymbol{y} - \boldsymbol{z}) + \tfrac{\rho}{2}\|\boldsymbol{y} - \boldsymbol{z}\|^2$$

Then, expanding the ADMM updates introduced in Equation (6) yields:

$$\boldsymbol{y}^{k+1} = \arg\min_{\boldsymbol{y}} F(\boldsymbol{y}) + \tfrac{\rho^k}{2}\|\boldsymbol{y} - \boldsymbol{z}^k + \boldsymbol{u}^k\|^2$$

$$\boldsymbol{z}^{k+1} = \arg\min_{\boldsymbol{z}} G(\boldsymbol{z}) + \tfrac{\rho^k}{2}\|\boldsymbol{y}^{k+1} - \boldsymbol{z} + \boldsymbol{u}^k\|^2$$

$$\boldsymbol{u}^{k+1} = \boldsymbol{u}^k + \boldsymbol{y}^{k+1} - \boldsymbol{z}^{k+1}$$

These updates are equivalent to the proximal splitting form presented in Section 5, with $\mathrm{prox}_F$ realized by the $\boldsymbol{y}$-update and $\mathrm{prox}_G$ by the $\boldsymbol{z}$-update. The dual variable $\boldsymbol{u}$ accumulates the residual mismatch between the two copies, driving them toward consensus. At convergence, $\boldsymbol{y} = \boldsymbol{z}$, recovering the minimizer of $F + G$.

## H  COMPLETE PROOFS

**Proof of Theorem 6.1.** *Claim:* $\mathrm{dist}_\mathcal{C}(\tilde{\boldsymbol{x}}_0^t) \le \frac{1}{\sqrt{\lambda_t \eta_t}}\mathrm{dist}_\mathcal{C}(\hat{\boldsymbol{x}}_0^t)$

**Lemma H.1.** *Let* $\phi(\boldsymbol{z}) = \frac{1}{2\eta_t}\|\boldsymbol{z} - \hat{\boldsymbol{x}}_0^t\|^2 + \frac{\lambda_t}{2}\mathrm{dist}_\mathcal{C}(\boldsymbol{z})^2$. *By optimality of* $\hat{\boldsymbol{x}}_0^t$, $\phi(\tilde{\boldsymbol{x}}_0^t) \le \phi\big(\Pi_\mathcal{C}(\hat{\boldsymbol{x}}_0^t)\big)$.

Lemma H.1 holds as the minimizer's objective value can't exceed the value at any feasible point, in particular at the projection of $\hat{\boldsymbol{x}}_0^t$.

Then, expanding from the lemma:

$$\frac{1}{2\eta_t}\|\tilde{\boldsymbol{x}}_0^t - \hat{\boldsymbol{x}}_0^t\| + \frac{\lambda_t}{2}\mathrm{dist}_\mathcal{C}(\tilde{\boldsymbol{x}}_0^t)^2 \le \frac{1}{2\eta_t}\|\Pi_\mathcal{C}(\hat{\boldsymbol{x}}_0^t) - \hat{\boldsymbol{x}}_0^t\| + \frac{\lambda_t}{2}\mathrm{dist}_\mathcal{C}\big(\Pi_\mathcal{C}(\hat{\boldsymbol{x}}_0^t)\big)^2$$

Since $\frac{\lambda_t}{2}\mathrm{dist}_\mathcal{C}\big(\Pi_\mathcal{C}(\hat{\boldsymbol{x}}_0^t)\big)^2 = 0$, this term can then be omitted.

$$\frac{1}{2\eta_t}\|\tilde{\boldsymbol{x}}_0^t - \hat{\boldsymbol{x}}_0^t\| + \frac{\lambda_t}{2}\mathrm{dist}_\mathcal{C}(\tilde{\boldsymbol{x}}_0^t)^2 \le \frac{1}{2\eta_t}\|\Pi_\mathcal{C}(\hat{\boldsymbol{x}}_0^t) - \hat{\boldsymbol{x}}_0^t\|$$

Dropping the non-negative first term gives:

$$\frac{\lambda_t}{2}\mathrm{dist}_\mathcal{C}(\tilde{\boldsymbol{x}}_0^t)^2 \le \frac{1}{2\eta_t}\|\Pi_\mathcal{C}(\hat{\boldsymbol{x}}_0^t) - \hat{\boldsymbol{x}}_0^t\|$$

or equivalently

$$\boxed{\mathrm{dist}_\mathcal{C}(\tilde{\boldsymbol{x}}_0^t) \le \frac{1}{\sqrt{\lambda_t \eta_t}}\mathrm{dist}_\mathcal{C}(\hat{\boldsymbol{x}}_0^t)}$$

$\square$

**Proof of Theorem 6.2.** *Claim:* $\mathbb{E}\big[\mathrm{dist}_\mathcal{C}(\boldsymbol{x}_0)^2\big] \le \frac{K}{2c_1}$

Begin by assuming $\mathbb{E}\big[\mathrm{dist}_\mathcal{C}(\hat{\boldsymbol{x}}_0^t)^2\big] \le K$, and the schedule is defined such that $\lambda_t = \frac{c_t}{\eta_t}$.

By Theorem 6.1, it is known that

$$\mathrm{dist}_\mathcal{C}(\tilde{\boldsymbol{x}}_0^t) \le \frac{1}{\sqrt{2\lambda_t \eta_t}}\mathrm{dist}_\mathcal{C}(\hat{\boldsymbol{x}}_0^t)$$

and thus, on the squared expectation

$$\mathbb{E}\big[\mathrm{dist}_\mathcal{C}(\tilde{\boldsymbol{x}}_0^t)^2\big] \le \frac{1}{2\lambda_t \eta_t}\mathbb{E}\big[\mathrm{dist}_\mathcal{C}(\hat{\boldsymbol{x}}_0^t)^2\big].$$

Then, by our assumption,

$$\mathbb{E}\big[\text{dist}_{\mathcal{C}}(\tilde{\boldsymbol{x}}_0^t)^2\big] \leq \frac{1}{2\lambda_t \eta_t} K = \frac{K}{2\lambda_t \eta_t}$$

Substituting our schedule $\lambda_t = \frac{c_t}{\eta_t}$ yields:

$$\mathbb{E}\big[\text{dist}_{\mathcal{C}}(\tilde{\boldsymbol{x}}_0^t)^2\big] \leq \frac{K}{2c_t}$$

By applying Corollary 6.2 at time $t = 1$:

$$\boxed{\mathbb{E}\big[\text{dist}_{\mathcal{C}}(\boldsymbol{x}_0)^2\big] \leq \frac{K}{2c_1}}$$

Thus, if the schedule of $c_t$ is decreasing, the expected violation converges as $t \to 0$.

$\square$

***Proof of Theorem 6.3.*** *Claim:* Existence of a local solution for the proximal mapping.

*Existence:* $\text{prox}_{\eta_t, g}$ *is continuous and coercive, hence attains a global minimizer for every $\hat{\boldsymbol{x}}_0^t$. In particular,* $\arg\min \text{prox}_{\eta_t, g} \neq \varnothing$.

First, assume $\eta_t, \lambda_t > 0$, $\mathcal{C} \subset \mathbb{R}^{3 \times n}$ nonempty, closed. For ease of notation:

$$\Phi_t(\boldsymbol{x}) = \frac{1}{2\eta_t}\|\boldsymbol{x} - \hat{\boldsymbol{x}}_0^t\|^2 + \frac{\lambda_t}{2}\text{dist}_{\mathcal{C}}(\boldsymbol{x})^2$$

Since the distance from the constraint set is strictly non-negative,

$$\Phi_t(\boldsymbol{x}) \geq \frac{1}{2\eta_t}\|\boldsymbol{x} - \hat{\boldsymbol{x}}_0^t\|^2 = \frac{1}{2\eta_t}\|\boldsymbol{x}\|^2 + \frac{1}{2\eta_t}\langle \boldsymbol{x}, \hat{\boldsymbol{x}}_0^t\rangle + \frac{1}{2\eta_t}\|\hat{\boldsymbol{x}}_0^t\|^2.$$

Since the term $\|\boldsymbol{x}\| \to \infty$ as $\boldsymbol{x} \to \infty$, $\Phi$ is $\Phi$.

$\Phi$ is also lower semicontinuous since (i) the quadratic anchor term $\frac{1}{2\eta_t}\|\boldsymbol{x} - \hat{\boldsymbol{x}}_0^t\|^2$ is continuous, and (ii) the squared distance to the closed set is lower semicontinuous and finite everywhere.

Since $\Phi$ is coercive and lower semicontinuous, it obtains a global minimum.

$$\boxed{\arg\min \Phi_t \neq \varnothing}$$

*Local uniqueness.* *If $\hat{\boldsymbol{x}}_0^t$ lies within the prox-regularity neighborhood of $\mathcal{C}$, then the projection $\Pi_{\mathcal{C}}(\hat{\boldsymbol{x}}_0^t)$ is single-valued. Moreover, if $\text{prox}_{\eta_t, g}$ is strongly convex in a neighborhood of $\Pi_{\mathcal{C}}(\hat{\boldsymbol{x}}_0^t)$, then the proximal minimizer is unique within that neighborhood.*

Now, assume $\mathcal{C}$ is prox-regular at all points in a neighborhood $\mathcal{X}$ of $\boldsymbol{x}^\star = \Pi_{\mathcal{C}}(\hat{\boldsymbol{x}}_0^t)$. Prox-regularity then implies:

- **Single valued projection.** The projection $\Pi_{\mathcal{C}}(\cdot)$ is single valued and Lipschitz on the neighborhood $\mathcal{U} \subset \mathcal{X}$.

- **Smooth distance function.** Thus, $\text{dist}_{\mathcal{C}}(\cdot)^2$ is $C^1$ on $\mathcal{U}$ with gradient:

$$\nabla\left(\tfrac{1}{2}\text{dist}_{\mathcal{C}}(\boldsymbol{x})^2\right) = \boldsymbol{x} - \Pi_{\mathcal{C}}(\boldsymbol{x}), \qquad \boldsymbol{x} \in \mathcal{U}.$$

Let $\mathcal{U}$ be small enough that the above holds and $\hat{\boldsymbol{x}}_0^t \in \mathcal{U}$. Consider $\Phi_t$ restricted to $\mathcal{U}$. Its gradient is

$$\nabla\Phi_t(\boldsymbol{x}) = \frac{1}{\eta_t}(\boldsymbol{x} - \hat{\boldsymbol{x}}_0^t) + \lambda_t(\boldsymbol{x} - \Pi_{\mathcal{C}}(\boldsymbol{x})), \qquad \boldsymbol{x} \in \mathcal{U},$$

and its (generalized) Hessian can be bounded below using the Jacobian of $\Pi_{\mathcal{C}}$. In particular, since $\Pi_{\mathcal{C}}$ is nonexpansive in $\mathcal{U}$,

$$\langle \nabla^2\Phi_t(\boldsymbol{x})\,v, v\rangle \geq \frac{1}{\eta_t}\|v\|^2 + \lambda_t\langle(I - D\Pi_{\mathcal{C}}(\boldsymbol{x}))v, v\rangle \geq \frac{1}{\eta_t}\|v\|^2,$$

for all $v$ and almost every $\boldsymbol{x} \in \mathcal{U}$. Thus $\Phi_t$ is locally $\mu$-strongly convex on $\mathcal{U}$ with $\mu = 1/\eta_t > 0$.

Local strong convexity implies a unique critical point in $\mathcal{U}$, hence a unique local minimizer of $\Phi_t$ in $\mathcal{U}$. **Since $\Phi_t$ is coercive with at least one global minimizer, and $\hat{x}_0^t$ is in the prox-regularity neighborhood so the minimizer lies in $\mathcal{U}$, the proximal minimizer is unique in that neighborhood.**

$\square$

