# OpenReview forum: "Constrained Diffusion for Protein Design with Hard Structural Constraints"
_ICLR.cc/2026/Conference — ICLR 2026 Poster_

### Official Review · Reviewer_dnLp · 2025-11-01

**Soundness:** 3
**Presentation:** 3
**Contribution:** 3
**Rating:** 8
**Confidence:** 5

**Summary:**

This author proposed a method to solve a gap in protein backbone generation: enforcing hard functional and stereochemical constraints during diffusion-based design. The authors rproposed emthod use reverse diffusion with a predict–prox–renoise procedure: (i) predict a clean structure, (ii) apply a proximal feasibility update (with a Moreau–envelope penalty that tightens over time), and (iii) re-noise to stay on the data manifold. They further decouple local stereochemistry from global topology via a consensus ADMM split, enabling separate proximal updates that are warm-started across steps. Theoretical results bound constraint violations and motivate schedules for the penalty and trust parameters. Empirically, the method is evaluated on (1) a new PDZ motif-scaffolding benchmark and (2) vacancy-constrained pocket design with box/void geometry. Results claim perfect constraint satisfaction and strong usable rates vs. RFdiffusion baselines and constraint-guided SMC, while maintaining structural diversity and reasonable compactness.

**Strengths:**

The predict–prox–renoise design is simple, modular, and theoretically grounded; the scheduling guidance (λ↑ as σ↓; η tied to diffusion variance) is clear and actionable. ADMM splitting of local vs. global constraints is well-motivated for proteins (stereochemistry vs. long-range couplings) and yields a practical proximal scheme with consensus guarantees under mild conditions. Theorems quantify contraction of constraint violations and give scheduling rules that drive terminal feasibility, aligning with the method’s design. On the PDZ benchmark, baselines produce no usable designs whereas the proposed method yields 21% usable (up to 83% for well-posed ligands) with perfect constraint satisfaction and better diversity/compactness; similarly strong wins appear in the vacancy-constraint task.

**Weaknesses:**

Feasibility guarantees rely on prox-regularity; highly nonconvex sets common in protein design may violate this, limiting formal guarantees. Proximal/ADMM steps add overhead and may require careful tuning of λ/η/ρ schedules; guidance on sensitivity and robustness across architectures and tasks could be expanded. Results are strong on PDZ and a geometric vacancy constraint, but broader functional constraints (e.g., catalytic geometry, interface polarity, multi-chain assemblies) and end-to-end sequence design with wet-lab validation remain future work.

**Questions:**

1. Beyond PDZ scaffolding and geometric vacancy control, which biologically grounded constraints (e.g., catalytic triad geometry, specific H-bond networks) most motivate hard-constraint enforcement, and how will the framework extend to them?
2. Can you quantify how often unconstrained RFdiffusion fails due to global vs. local constraint violations in real pipelines, to better motivate where hard constraints matter most?
3. How does enforcing exact feasibility impact downstream functional success (e.g., binding affinity predictions) compared with soft guidance—any surrogate metrics supporting the motivation?
4. The Moreau-envelope penalty transitions toward hard constraints as λ→∞. How sensitive are outcomes to the λt and ηt schedules, and do you provide heuristics or automatic tuning beyond the variance-linked choice?
5. In the ADMM split, how do you select which residues belong to the “local” vs. “global” blocks in more complex topologies, and what is the computational cost per sweep as structure size grows?
6. On the PDZ benchmark, your method attains 21% usable overall (up to 83% for favorable cases). What characteristics define “well-posed” ligands, and how general is that 83% across targets?
7. In the vacancy-constraint experiment, could you provide additional quality metrics (e.g., secondary-structure preservation per segment, clash scores) to complement constraint satisfaction and radius of gyration?

---

> ### Author Response · Authors · 2025-11-20
>
> Thank you for the time you dedicated to this detailed and constructive review! We appreciate that the strengths of our work were identified and highlighted, and we find all the comments helpful for further improving it. We address each of the points in detail below.
>
> > **W1: Prox-regularity Assumption for Theoretical Results**
>
> Indeed, we agree that the feasiblity guarantees do not extend generally to nonconvex constraint sets. However, we will emphasize that theoretical guarantees are typically derived for convex constraint sets, and we consider the extension of our results to prox-regular constraint sets a significant contribution. Additionally, please note that prox-regularity applies almost always for the nonconvex constraint sets considered in this work. In the case of Section 7.2, for example, the projection is unique almost everywhere, with the exception of the cones apex and edges; hence, prox-regularity applies at almost all points, and a similar argument can be made for the experiments in Section 7.1.
>
> Additionally, as the reviewer is surely aware, general guarantees for nonconvex constraints remain unobtainable within the optimization field as a whole. Hence, _our guarnatees for prox-regular sets are among the strongest attainable via convex optimization theory_.
>
> > **Q1 and W3: Additional Validation for Other Constraints and Sequence Design**
>
> We completely agree that additional evaluation on other problem settings is a compelling direction for future work! We would also like to emphasize that the results in this paper provide a strong starting point for these evaluations, due to the generality of the constraints explored. The two experiments in Section 7.1 and 7.2 are indeed chosen as *representative instances of much broader families of constraints*, rather than isolated cases studies. In particular:
>
> 1. The constraints used in *Section 7.1 generalize to other binding tasks.* These experiments are defined on residue-residue interactions, specifically controlling spatial relationships in defined regions. These exact constraints are relevant for a broad array of tasks, including protein-protein binder engineering, interface resdesign, and cofactor binding sites, to name a few. The only aspect that would change from one setting to another is which residues are being constrained, which our proposed method handles natively. Hence, the implications of our evaluation in 7.1, indeed extend to wide array of similar protein engineering tasks. We are currently engaged in internal collaborations which extend this framework to several of these similar tasks, well beyond the PDZ domain.
>
> 2. The constraints used in *Section 7.2 generalize to geometric designs.*  These experiments are certainly viable for pocket design, but they are also viable for many other tasks. For instance, they are applicable to binding tasks where hot-spots may not be directly known, but where a region must adopt a targeted geometric configuration (e.g., polar versus hydrophobic surface matching).  Furthermore, settings such as cavity engineering and coiled-coil design could similarly exploit identical constraint types. Thus, the constraints explored in this experiment are likely the exact same as would be used in a variety of other tasks.
>
> Furthermore, *note that our algorithm never encodes problem-specific heuristics*, but operates by imposing differentiable constraints on generic design properties. We belive that illustrating that our approach can capture these general types of constraints opens up a wide-array of applications. While these additional settings are beyond the scope of a conference length ML paper, very likely consituting targeted application studies in their own right, we argue that the evaluation
> effectively covers the *types of constraints* applicable to many important protein engineering applications.

---

> ### Author Response · Authors · 2025-11-20
>
> > **Q2: Failure Cases for Vanilla RFdiffusion**
>
> The standard implementation of RFdiffusion excels at modeling backbones which satisfy stereochemical properties. Local constraint violations are very rare in standard backbone structure diffusion models, with many explicitly encoding these properties into the diffused representation. Rather, the challenge with these unconstrained implementations is that they are very difficult to control (e.g., enforcing global constraints on functional properties). For many specific functional requirements such as the $\beta$-sheet bindings in Section 7.1, without explicit constraints RFdiffusion _will never exactly satisfy the required criteria_. We posit this as an out-of-distribution phenomenon that generative models alone cannot overcome natively, not even with guidance-based methods.
>
> On the other hand, local constraint violations tend to arise solely when global constraints are explicitly enforced. This occurs as enforcing the global constraint causes the associated residues to shift, requiring that neighboring residue positions are adjusted to maintain local constraint adherence. Hence, in cases where global constraints are not required, unconstrained RFdiffusion is sufficient and local constraint enforcement is unnecessary.
>
>
> > **Q3: Impact of Constraints on Downstream Success**
>
> We appreciate this thoughtful question. Assessing how exact geometric feasibility influences downstream functional performance is indeed an important direction for future work. In principle, enforcing strict feasibility should reduce structural ambiguity and thereby improve sequence designability and functional predictions. One natural next step is to generate sequences with inverse-folding models, followed by AlphaFold2 structure prediction and evaluation with surrogate metrics such as pLDDT, pTM, pAE, and Rosetta energy terms. These provide interpretable proxies for backbone viability, packing quality, and potential binding affinity prior to wet-lab validation with our experimental collaborators.
> However, these sequence and function-level assessments in wet-lab are well beyond the scope of the present manuscript, which is focused specifically on the ML framework for generating backbones that satisfy complex constraints. We see downstream functional evaluation, both in silico and experimentally, as a natural extension of this work, and we look forward to reporting those results in near future studies. Thank you again for the comment!
>
> > **Q4, W2: Expanded Discussion of Hyperparameter Scheduling**
>
> Thank you for the suggestion! While running the evaluation included in the submission, we did not invest substantial effort into tuning these hyperparameters, but we agree that it would strengthen the paper if empirical evidence was provided to assure readers of the robustness to different hyperparameter setups. During the rebuttal period we conducted additional ablation for the $\lambda$ schedule, which we provide below.
>
> |                             | $\lambda = \frac{10}{t}$ | $\lambda = \frac{100}{t}$ | $\lambda = \frac{1000}{t}$ | $\lambda \to \infty$ |
> |-----------------------------|--------------------------|---------------------------|----------------------------|-------------------------|
> | Constraint Satisfaction (%) | **100.0**                | **100.0**                 | **100.0**                  | **100.0**               |
> | Structure Realism (%)       |  80.0                    | **100.0**                 | **100.0**                  |  90.0                   |
> | Usable Percentage (%)       |  80.0                    | **100.0**                 | **100.0**                  |  90.0                   |
> | Diversity (%)               |  80.0                    | **100.0**                 | **100.0**                  |  90.0                   |
> | Radius of Gyration (A)      | 10.58                    | 10.53                     | 10.56                      | **10.34**               |
>
> We perform this ablation of $\lambda$ on the PDZ design $\mathrm{6ubh}$ from our benchmarking set, a well-posed structure ideal for this constrained generation application (see our response to Q6 for more details on this selection criteria). These results are indeed interesting; stricter constraint enforcement at earlier timesteps (e.g., higher $\lambda_T$) provides more physically realistic outputs as the samples are more strongly biased towards the feasible subdistribution. However, increasing this too strictly, results in over constraining early states, analogous to placing an overly large weight on the diffusion-guidance terms as discussed in Section 3.
>
> We hope that these additional results have answered your question! We intend to add detailed versions of this ablations to the paper's appendix and have already added these table to the revised paper submission.

---

> ### Author Response · Authors · 2025-11-20
>
> > **Q5: Local vs. Global Blocks and Runtime Scaling**
>
> Thank you for the question. In short, the local constraints will always apply to all residues, but the global constraints often only apply to a subset.
>
> To be more specific, all residues will be subject to the local constraints (e.g., stereochemical properties), so the entire structure will be considered when applying the proximal update $\text{prox}\_{\rho^k, F}$.
> Conversely, the global constraints may act on a subset of the residues that are subject to the application-specific functional constraints. While the proximal update $\text{prox}\_{\rho^k, G}$ is applied to a complete representation of the backbone, as $z$ is a copy of the predicted backbone $\hat{x}^t_0$, the constraints and proximal operator often only act on specific residues. In Section 7.1, this acts on up to five residues, subject to the peptide's available backbone hydrogen bond donors and acceptors, as visualized in the yellow $\beta$-sheet from Figure 2. However, in Section 7.2, the entire backbone is subject to the global constraints, as all residues must fall inside the box yet outside the exclusion zone; hence, in this case, the proximal update $\textrm{prox}_{\rho^k, G}$ could modify any residue in the backbone.
>
> As far as runtime scaling, this is tied much more closely to the complexity of the constraints rather than the structure size. For instance, consider the global geometric constraints in Section 7.2, which are applied to *all residues in the structure* as oppposed to the three to five binding residues in Section 7.1. Despite imposing a larger number of constraints across the complete structure, the runtime remains an order of magnitude smaller than the runtime in Section 7.1. In this case, the projection operator converges much faster due to the simpler (yet stilll nonconvex) geometric constraints in question. Hence, the computational cost per sweep is largely dictated by the constraints themselves, and the scalability to larger structures will be primarily dictated by the constraints in question.
>
> > **Q6: Characteristics of Well-Posed Ligands**
>
> We agree with the reviewer that this sentence was ambiguous in the context of the preceding description. We have added clarifying details explaining our manual procedure for selecting more favorable PDZ constructs, particularly in cases involving very short peptide targets or peptides containing geometrically restrictive residues such as prolines.
>
> > **Q7: Additional Quality Metrics for Vacancy Constrained Setting**
>
> Thank you for the suggestion! We agree that the inclusion of additional structural metrics will help to strengthen our evaluation. During the rebuttal period we assessed the generated samples on the metrics suggested, which we will highlight below:
>
> |                                               | RFdiffusion | Recenter  | CGD       | Ours      |
> |-----------------------------------------------|-------------|-----------|-----------|-----------|
> | Percentage of Residues in Secondary Structure | 80.8%       | 82.5%     | 83.5%     | **85.0%** |
>
>
> Regarding clash scores, we note that this metric is not directly comparable in the vacancy-constraint setting; constrained methods necessarily place residues within a restricted spatial region, resulting in inflation of repulsive-energy-based clash metrics. As a result, these scores primarily reflect the geometry of the imposed constraint rather than differences between the methods. However, we believe the secondary structure preservation metrics are quite informative: _our method improves secondary structure conformation frquency as compared to the baselines, highlighting that the method effectively captures the constraints without degrading the generation quality._
>
> Again, thank you for the suggestion to add these results. We have added these to the updated submission of the paper in Appendix B!
>
>
> ---
> Thank you again for your review and for your positive assessment of our work. We believe our response has clarified all the outstanding questions, but we are happy to provide additional details as needed. Many thanks!

---

### Official Review · Reviewer_TyWr · 2025-11-01

**Soundness:** 2
**Presentation:** 3
**Contribution:** 2
**Rating:** 2
**Confidence:** 3

**Summary:**

This paper presents a constrained diffusion framework for protein design, aiming to enforce hard structural and functional constraints.1 The core method is a "predict-prox-renoise" stochastic proximal method, which applies feasibility corrections to the predicted clean state $\hat{x}_{0}^{t}$ rather than the noisy state $x_t$.1 The authors also propose an ADMM decomposition to decouple local and global constraints.1 The method is evaluated on two tasks: motif scaffolding (using a newly introduced PDZ benchmark) and vacancy-constrained pocket design, claiming 100% constraint satisfaction rates where baselines achieve 0%.

**Strengths:**

Novel and well-motivated "predict-prox-renoise" method for hard constraints.

(Superficially) perfect 100% constraint satisfaction on complex tasks.

Contributes a new, curated benchmark dataset for PDZ motif scaffolding.

**Weaknesses:**

Fails to cite or compare against any true SOTA competitors in motif scaffolding (e.g., Genie 2 , OriginFlow ), invalidating its SOTA claim.

The "Ours" method in Exp 2 is a combination of the new method and the baselines , making results impossible to attribute.

The ADMM method  is not general , and the theory  does not apply to the problem.

 0% success for all baselines is not credible.

**Questions:**

see weakness

---

> ### Author Response · Authors · 2025-11-20
>
> Thank you for the time invested in reviewing our work and, in particular, your acknowledgement of our methodology as novel and well-motivated. In the response below, we address the points raised.
>
>
> **W1: Combination of New Method and Baselines**
>
> Thank you for pointing this out. We agree that adding a baseline which combines Recentering and Constraint-Guided conditioning would improve the interpretability of the results. During the rebuttal period, we conducted additional evaluation to provide this comparison:
>
> |                             | Recenter + CGD | Ours      |
> |-----------------------------|----------------|-----------|
> | Constraint Satisfaction (%) | 27.4           | **100.0** |
> | Structure Realism (%)       | 93.8           | **97.8**  |
> | Usable Percentage (%)       | 24.2           | **97.8**  |
> | Diversity (%)               | 24.2           | **97.8**  |
> | Radius of Gyration (A)      | 26.6           | **14.8**  |
>
> This baseline, indeed, performs slightly better than the isolated Constraint-Guided conditioning included previously. The constraint satisfaction increases from 21.6% to 27.4%, and the useable percentage improves from 20.5% to 24.2%. While this outperforms Constraint-Guided conditioning marginally, the takeaway is clear: *the proximal updates proposed by our constrained diffusion method dramatically improve the constraint satisfaction and functional utility of the generations.*
>
> We hope that these additional results have helped assure you that the improved performance provided by our method can be attributed to the contributions of this work. We appreciate the suggestion to include these results, and we have added these to the updated submission of the paper in Section 7.2!
>
> **W2: Comparison to Other Generative Models**
>
> First, we do appreciate you calling our attention to models such as Genie 2 and OriginFlow; we have extended the discussion of related work in Section 8 to include these models and emphasize the growing interest in diffusion (and flow matching) for protein design and we invite you to check the revised version of the paper.
>
> It is important however to clarify that _our approach does not introduce a competing diffusion model, but rather a general framework applicable to any diffusion process_. Specifically, our approch operates orthogonally to the underlying generative model, and _it can be integrated into any other backbone structure diffusion models_ (as mentioned in lines 295-297) including Genie 2 and OriginFlow.
>
> Hence, the goal of our evaluation is not to demonstrate that one particular generative model is better than another, but instead we aim to show that our method improves over the underlying generative process. In that sense, our methodology is model-agnostic, and the inclusion of such comparisons would not further evaluate our method. Rather, methods like Genie 2 and OriginFlow are **separate lines of research focused on model design**, whereas our work presents a general mechanism for integrating formal constraints into an arbitrary underlying diffusion model.
>
> Additionally, we believe it is worth noting that while both of the explicitly referenced models appear promising, to the best of our knowledge, *they are yet to successfully pass the peer review process*. At the time of this study, RFdiffusion remains an established and mature tool for de novo design, making it a natural choice for our evaluation.
>
>
> [1/2]

---

> > ### Author Response · Authors · 2025-11-20
> >
> > **W3: Generality of ADMM and Theoretical Results**
> >
> > We respectfully disagree with the claim that ADMM is "not general". ADMM is a widely used and theoretically grounded optimization framework, which is applicable to a broad array of constrained optimization problems, including both convex and nonconvex settings. Our use of ADMM is not tied to a specific set of design constraints, but rather has been presented in a general operator-splitting form. Indeed, $g_\mathrm{local}$ is present in any protein design task (e.g., stereochemical properties), and $g_\mathrm{global}$ is present in exactly the types of problems that this paper is written to address -- those with functional constraints such as in Section 7.1 and 7.2. Given that **ADMM is a foundational and extensively validated method within constrained optimization**, we believe this claim is misleading. Perhaps we are misinterpreting your point? Can the reviewer substantiate their claim with specific reasoning?
> >
> > We similarly argue that the claim "the theory does not apply to the problem" mischaracterizes the theoretical results derived. While the evaluation explores nonconvex constraints, these **constraint sets are indeed prox-regular at almost all points**. Prox-regularity holds if for each viable point and normal direction, small perturbations still project uniquely and smoothly back to $\mathbf{C}$. In the case of Section 7.2, for example, the projection is unique almost everywhere, with the exception of the cones apex and edges; hence, prox-regularity applies at almost all points for Section 7.2, and a similar argument can be made for Section 7.1.
> >
> > As the reviewer will surely know, guarantees for general nonconvex constraints remain out of reach for the optimization field at large, so proving results under prox‑regularity *should be considered a strength, and not a limitation!* Likewise, our method's empirical success for nonconvex constraint sets should be viewed as a strength of our algorithmic contributions, highlighting the efficacy of the ADMM decomposition for constrained generative modeling tasks.
> >
> >
> > **W4: Credibility of Results**
> >
> > While we acknowledge the reviewer's concern, we'd like to assure the reviewer that this behavior is actually expected. As discussed in Section 8 (lines 452-458), current paradigms rely on sampling tens of thousands of candidates in hopes of generating a few viable designs. In our settings, where exact constraint satisfaction is necessary over several binding points, 0% success rate is certainly in line with the expected outcome, and this exact outcome is what motivated the study.
> >
> > Additionally, we stress that **the code and benchmark were submitted**, allowing full reproduction of the results reported in the paper. We invited the reviewer to run them. Should the reviewer argue this point as grounds for their low scores, we strongly compel them to **verify the results using the provided materials before making such a determination**, as all experiments are fully reproducible and transparently documented.
> >
> > ---
> >
> > Thank you, again, for your feedback. We believe our response has addressed your concerns, providing additional explanations and results. Given your acknowledgement of our novel method and benchmark, we hope that you would reconsider your assessment in light of these clarifications. We are happy to follow-up if any questions remain. Many thanks!
> >
> > [2/2]

---

> ### Author Response · Authors · 2025-11-26
> **Follow-up and clarification**
>
> As the discussion period is nearing its end, we wanted to ask **Reviewer TyWr** if there are any follow-up points we can clarify.
>
> We believe we have responded to all questions and concerns raised, in addition to taking a few days to run the additional experiments necessary to demonstrate our points, all of which are incorporated in the revised version of the paper.
> \
> If there are no further points of clarification regarding the manuscript, we kindly ask that reviewer TyWr consider increasing their score to reflect the improvements and clarifications we have provided. We are happy to continue to engage in discussion and answer any questions!

---

### Official Review · Reviewer_ausW · 2025-11-02

**Soundness:** 3
**Presentation:** 2
**Contribution:** 3
**Rating:** 6
**Confidence:** 3

**Summary:**

This paper presents a constrained diffusion framework for precision engineering of detailed molecular features in proteins. Authors validate their approach in silico on a contributed tasks and novel curated benchmark datasets.

**Strengths:**

- This is definitely an interesting and strong contribution focusing on a slightly neglected aspect that is however of very high practical & theoretical relevance in precision engineering of protein architectures & binding sites.
- The proposed solution is benchmarked against convincing baselines and authors report significant improvements.
- Code & benchmark data is released.

**Weaknesses:**

- I feel a benchmark is scarce, it’s a relevant problem but I would welcome an effort to introduce more interesting cases. There are many problems requiring precision engineering that could show that the method is robust to different cases. E.g. one widely recognised problem used to study precision protein engineering for long time is parametric protein design with coiled coil bundles.
- While the technical / implementation aspects of the method are properly described I think the formulation and description of the proposed benchmark dataset is less understandable (I suppose even for struct/bio readearship, not mentioning purely ML audience). I suggest to rewrite this section so it reads more clear (also minor thing - the PBM is never defined in the paper).
- Finally, I believe being able to come up with backbones that satisfy the complex geometric criteria is an important contribution I have one follow up q to the authors - are these backbones designable? I miss the experiment that would prove (in silico would naturally be enough given the venue) that indeed we can design compatible sequences that will fold into structures that are predicted by the authors e.g. with the ProteinMPNN / AF2 pipeline that authors already set up.

**Questions:**

See weaknesses for potential discussion points.

---

> ### Author Response · Authors · 2025-11-20
>
> Let us begin by expressing our appreciation for your thoughtful assessment; we especially appreciate your acknowledgement of the high practicality of our work for protein design pipelines and your focus on the significance of this study within real biological applications. We address your outstanding concerns below.
>
>  **W1: Additional Applications**
>
> Thank you for this suggestion; we agree that it is important to provide an evaluation which captures the robustness of the method across different problem sets, and we will subsequently argue that our evaluation indeed provides this.
>
> First, the goal of our approach has been to propose a *general* method for enforcing fine-grained design constraints for protein structures. The two experiments in Section 7.1 and 7.2 are chosen as *representative instances of much broader families of constraints*, rather than isolated cases studies.
>
> 1. **Section 7.1 constraints generalize to other binding tasks.** The constraints used in this experiment are defined on residue-residue interactions, specifically controlling spatial relationships in defined regions. These exact constraints are relevant for a broad array of tasks, including protein-protein binder engineering, interface redesign, and cofactor binding sites, to name a few. The only aspect that would change from one setting to another is which residues are being constrained. Hence, the implications of our evaluation in 7.1, indeed extend to wide array of similar protein engineering tasks. In fact, we are currently engaged in internal collaborations which extend this framework to several of these similar tasks, well beyond the PDZ domain.
>
> 2. **Section 7.2 constraints generalize to geometric designs.** The constraints used in this experiment are practical for pocket design as shown, but they are also viable for many other tasks. For instance, these are applicable to binding tasks where hot-spots may not be explicitly known, but where a specific region of a structure is being targeted (e.g., binding to specific polar or hydrophobic surfaces). Furthermore, settings such as cavity engineering and coiled-coil design could similarly exploit identical constraint types (e.g., cylindrical displacements). Thus, the constraints explored in this experiment are the exact same as would be used in a variety of other tasks.
>
> Next, **note that our algorithm never encodes problem-specific heuristics**, but operates by imposing differentiable constraints on generic design properties. We belive that illustrating that our approach can capture these general types of constraints opens up a wide-array of applications. While these additional settings are beyond the scope of a conference length ML paper, very likely consituting targeted application studies in their own right, we argue that the evaluation
> effectively covers the *types of constraints* applicable to many important protein engineering applications.
>
> Finally, as you acknowledge, the introduction of this PDZ benchmark is a valuable step, which we believe will help to provide more unified comparisons for constrained protein engineering methods. At the time of this submission, widely adopted benchmarks for constrained protein design tasks are yet to be adopted, and this absence motivated our contribution of the PDZ benchmark. Furthermore, the benchmark was designed to provide this representative case, as described above, providing a setting with well-characterized, generalizable constraints. Thus, our efforts focused on this setting and on establishing an evaluation set that both reproducible and encourages the development of similar benchmarks for other applications within protein engineering.
>
>
> **W2: Description of the Proposed Benchmark**
>
> We appreciate the suggestion and agree that the exposition could be improved. We have revised the main text to improve accessibility for both biological and machine-learning audiences, using more consistent terminology and a clearer task description. Please see the updated manuscript, and we would be happy to address any additional questions.
>
>
> **W3: Backbone Design Results**
>
> We agree that demonstrating sequence designability, via inverse-folding models followed by structure prediction, represents an important next step for validating the practical utility of our generated backbones. We are actively pursuing this in parallel work by generating sequences with ProteinMPNN and evaluating their foldability using AlphaFold2, with the goal of experimentally testing selected designs in collaboration with our wet-lab partners. However, this direction is beyond the scope of the present manuscript, which focuses the ML methodology for generating backbones that satisfy complex constraints. We view designability assessment as a natural extension of this work, and we look forward to reporting those results in future follow-up studies!

---

> > ### Author Response · Authors · 2025-11-20
> >
> > ---
> > Thank you, again, for your time and feedback. We believe that the additional results and explanations included in our response have addressed the concerns presented, but we encourage you to follow-up on any points that remain unclear. Additionally, we encourage the reviewer to check the updated version of our submission, which has been extended to incorporate these additional results and discussions. We hope these revision will merit your strong support for this work!

---

> > > ### Comment · Reviewer_ausW · 2025-11-25
> > >
> > > I'd like to thank the authors for convincing rebuttal and provided changes & explanations. Taking this into the account, along with comments & responses from/to other reviewers, I believe this is a strong and important contribution and therefore I find it appropriate to increase my scoring.

---

> > > > ### Author Response · Authors · 2025-11-25
> > > >
> > > > Thank you very much again for your time and feedback. We remain available in case further questions arise.

---

### Official Review · Reviewer_muax · 2025-11-10

**Soundness:** 3
**Presentation:** 4
**Contribution:** 3
**Rating:** 6
**Confidence:** 3

**Summary:**

This paper introduces a framework for incorporating hard structural constraints into diffusion-based protein design. The authors identify that existing guidance methods often fail to satisfy precise, non-convex constraints required for functional design. They propose viewing the reverse diffusion process through the lens of proximal optimization. Their empirical results on PDZ scaffolding and molecule encapsulation show state-of-the-art constraint satisfaction (near 100% vs. about 0% for the baselines).

**Strengths:**

The overall idea of the proposed method is creative and explores a new direction for diffusion models. I see the main strengths of the paper as follows:
- strong results: the 100% satisfaction on PDZ vs 0% for strong baselines appears meaningful
- method: framing the reverse step as proximal optimization is a neat idea, and applying the correction to the estimated $\hat{x}_0$ instead of the noisy $x_t$ is well-motivated
- benchmark: the PDZ benchmark designed for the evaluation seems like a meaningful contribution in its own right

**Weaknesses:**

I believe there are a few minor weaknesses:
- baselines: while RFDiffusion is a good baseline, the authors should compare to the method in Christopher et al. (2024) [1] given the similarity between the proposed methods.
- the paper does not present any sensitivity studies for hyperparameters. How sensitive is the proposed approach to the schedule $\lambda$, the number of ADMM iterations, etc.?

[1]  Christopher et al., Constrained Synthesis with Projected Diffusion Models, 2024.

**Questions:**

- How sensitive is the proposed approach to the schedule $\lambda$, the number of iterations, etc.?
- How does the proposed method perform compared to the method in Christopher et al. (2024)?
- Did you try a baseline projecting $x_t$ directly? Did it fail as expected?

---

> ### Author Response · Authors · 2025-11-20
>
> Thank you for your thoughtful review and, in particular, for your recognition of our contributions from both a methodological standpoint and the benchmark we introduce. We reply to your main points below.
>
> **W1, Q2, Q3: Comparison to Christopher et al. (2024)**
>
> Thank you for this question, as it provides us an opportunity to provide additional results and discussion complimentary to the arguments presented in Section 3. Indeed, our methodology was inspired by the techniques introduced by Christopher et al. (2024), and our early experiments attempted to apply their approach for simple $\beta$-sheet pairings (e.g., binding between two beta-sheets with each being composed of four residues). Our observations illustrated that this approach was not viable for our domain, as a projection onto the local and global constraints failed to converge, especially at early noise levels. Stronger convergence was realized when projecting onto just the global constraints at later diffusion steps, but the realized structures never respected the local constraints in these cases, as we illustrate next.
>
> During the rebuttal period, we also applied Christopher et al. (2024) to the vacancy constraints in Section 7.2. We observed similar limitations to what described above; when projecting onto both local and global constraint sets, the Lagrangian method implemented for the projection operator (using code from the original paper) never converged. Greater success was observed when applying a closed-form projection onto the global constraints (e.g., consistent constraint satisfaction), but the projections similarly break local stereochemical constraints:
>
> |                             | Project $x_t$ (Local and Global) | Project $x_t$ (Global) | Ours      |
> |-----------------------------|----------------------------------|------------------------|-----------|
> | Constraint Satisfaction (%) | 0.0 (Failed to Converge)         | **100.0**              | **100.0** |
> | Structure Realism (%)       | N/A                              | 0.0                    | **97.8**  |
> | Usable Percentage (%)       | 0.0                              | 0.0                    | **97.8**  |
> | Diversity (%)               | N/A                              | N/A                    | **97.8**  |
> | Radius of Gyration (A)      | N/A                              | (13.1)                 | **14.8**  |
>
>
> These results reaffirm the limitations of integrating intermediate projection steps when working with high-dimensional data subject to complex nonconvex constraints. While in this setting it was possible to exactly satisfy the global constraints, the resulting structures are unusable, with high concentrations of atoms placed on the edges of the box and cone boundaries. When enforcing bot the local and global constraints explicitly, the projection operator never converged for any samples or timesteps. These results align closely with the intuitions discussed in Section 3, strongly motivating both clean state projections and the ADMM decomposition we introduce. We appreciate your suggestion to include these results, and we have included them in the revised submission.
>
> [1/2]

---

> > ### Author Response · Authors · 2025-11-20
> >
> > **W2, Q1: Hyperparameter Sensitivity Analysis**
> >
> > Thank you for the suggestion! While running the evaluation included in the submission, we did not invest substantial effort into tuning these hyperparameters, but we agree that it would strengthen the paper if empirical evidence was provided to assure readers of the robustness to different hyperparameter setups. During the rebuttal period we conducted this ablation for (1) $\lambda$ schedules and (2) ADMM iterations.
> >
> > |                             | $\lambda = \frac{10}{t}$ | $\lambda = \frac{100}{t}$ | $\lambda = \frac{1000}{t}$ | $\lambda = \inf$ |
> > |-----------------------------|--------------------------|---------------------------|----------------------------|-------------------------|
> > | Constraint Satisfaction (%) | **100.0**                | **100.0**                 | **100.0**                  | **100.0**               |
> > | Structure Realism (%)       |  80.0                    | **100.0**                 | **100.0**                  |  90.0                   |
> > | Usable Percentage (%)       |  80.0                    | **100.0**                 | **100.0**                  |  90.0                   |
> > | Diversity (%)               |  80.0                    | **100.0**                 | **100.0**                  |  90.0                   |
> > | Radius of Gyration (A)      | 10.58                    | 10.53                     | 10.56                      | **10.34**               |
> >
> > We perform this ablation of $\lambda$ on the PDZ design $\mathrm{6ubh}$ from our benchmarking set, a well-posed structure ideal for this constrained generation application (see our response to Reviewer dnLp Q6 for more details on this selection criteria). These results are indeed interesting; stricter constraint enforcement at earlier timesteps (e.g., higher $\lambda_T$) provides more physically realistic outputs as the samples are more strongly biased towards the feasible subdistribution. However, increasing this too strictly, results in over constraining early states, analogous to placing an overly large weight on the diffusion-guidance terms as discussed in Section 3.
> >
> > |                             | ADMM Iterations = 1 | ADMM Iterations = 2 | ADMM Iterations = 3 |
> > |-----------------------------|---------------------|---------------------|---------------------|
> > | Constraint Satisfaction (%) | **100.0**           | **100.0**           | **100.0**           |
> > | Structure Realism (%)       |  **80.0**           |  70.0               | 40.7                |
> > | Usable Percentage (%)       |  **80.0**           |  70.0               | 40.7                |
> > | Diversity (%)               |  **80.0**           |  70.0               | 40.7                |
> > | Radius of Gyration (A)      | **10.58**           | 10.72               | 11.24               |
> >
> > Next, we analyze the impact of additional ADMM iterations. As mentioned in Section 5, we find that it is only necessary to take a single sweep per diffusion step. In fact, the ablation suggests that multiple iterations degrade quality. The failure case that emerges over additional ADMM iterations is actually always tied to the formation of a $\beta$-sheet across the binding sites; additional iterations do not result in local stereochemical properties breaking, but the analysis suggests that they do lower the likelihood of secondary structure conformations. This is because a higher weight is placed on the ADMM optimization than the diffusion updates, leading to feasible generations but not ones which fall closely on the data manifold. Furthermore, we note that additional ADMM steps also become impractical because adding these iterations scales the runtime nearly linearly.
> >
> > We hope that these additional results have answered your questions! We intend to add detailed versions of both ablations here to the paper's appendix and have already added these tables to the revised paper submission in the appendix.
> >
> > ---
> >
> > Thank you, again, for your time and feedback. We believe that the additional results and explanations included in our response have addressed the concerns presented, but we encourage you to follow-up on any points that remain unclear. Additionally, we encourage the reviewer to check the updated version of our submission, which has been extended to incorporate these additional results and discussions. We hope these revision will merit your strong support for this work!
> >
> >
> > [2/2]

---

### Meta-Review · Area_Chair_tWRq · 2026-01-07

**Summary:**

This paper presents a constrained diffusion framework for protein design that integrates proximal feasibility updates and ADMM decomposition to enforce strict functional and stereochemical constraints during generation. The initial reviews were generally positive with three positive scores (6, 6, 8), and one distinct negative score (score 2). There is a consensus on methodological novelty of the "predict-prox-renoise" framework and the strong empirical results achieving near 100% constraint satisfaction where baselines (RFDiffusion) failed. Major concerns focused on (1) the lack of specific constrained-diffusion baselines (e.g., Christopher et al.), (2) hyperparameter sensitivity(muax, dnLp), and (3) limited scope in evaluation, scarce testing scenario, and lack of end-to-end design with downstream metrics such as designability (ausW, dnLp), (4) theoretical validity

**Reviewer Concerns:**

**Addressed concerns**:
- Missing Baselines (muax, TyWr): The authors provided comparisons to Christopher et al. (2024), showing it failed to converge or produced broken geometries, and added a "Recenter + CGD" baseline as requested by TyWr.
- Hyperparameter Sensitivity (muax, dnLp): New ablation studies on the $\lambda$ schedule and ADMM iterations demonstrated the method's robustness.
- Theoretical Validity (TyWr): The reviewer claimed ADMM is "not general" and the theory does not apply. The authors refuted this by citing standard optimization literature and clarifying that prox-regularity holds for their constraint sets.

**Partially Addressed**:
- Limited evaluation scope (ausW, dnLp): The authors explained how the PDZ and vacancy tasks represent broader classes of protein engineering problems, but leave more evaluation tasks as future work.
- Credibility of Results (TyWr): The negative reviewer simply stated, "0% success for all baselines is not credible." The authors noted that they have provided the code needed to reproduce the experiment, but the reviewer did not engage to retract the claim

**Outstanding**
- Downstream metrics (ausW, dnLp): The authors acknowledged that end-to-end design and wet-lab validation are future work, and did not provide downstream metrics such as designability, binding affinity as additional results. Reviewer ausW accepted this scope limitation.

**Reviewer Scores:**

Based on the rebuttal, I'd predict that reviewers with initial positive scores would maintain their rating,  reviewer TyWr might increase his rating if the code provided can be thoroughly checked and validated.

- Reviewer ausW (Original: 6): Actual: 8. Although the designability score is left as future work by the authors, this reviewer explicitly acknowledged the rebuttal addressed their concerns regarding benchmarks before the discussion closed.
- Reviewer dnLp (Original: 8): Predicted: 8. The reviewer was highly positive initially, and the rebuttal provided the requested secondary structure metrics and ablations.
- Reviewer muax (Original: 6): Predicted: 8. The authors provided the exact missing baseline (Christopher et al.) and the sensitivity analysis requested. The primary friction points for this reviewer were resolved and will likely increase their score.
- Reviewer TyWr (Original: 2): Predicted: 4. The Authors indicate that reviewer TyWr raised factually questionable claims (e.g., questioning the generality of ADMM) and misunderstood the scope of the work. The authors provided their code for reproducibility, and thus reviewer TyWr's concern regarding results credibility might be alleviated by examining the code.

---

### Decision · Program_Chairs · 2026-01-26

Accept (Poster)